# Adsorption of rare earth elements in regolith-hosted clay deposits

Anouk M. Borst [1]✉, Martin P. Smith [2], Adrian A. Finch [1], Guillaume Estrade [3], Cristina Villanova-de-Benavent [2], Peter Nason[2], Eva Marquis [2], Nicola J. Horsburgh [1], Kathryn M. Goodenough[4], Cheng Xu[5,6], Jindřich Kynický[7,8] & Kalotina Geraki [9]

Global resources of heavy Rare Earth Elements (REE) are dominantly sourced from Chinese regolith-hosted ion-adsorption deposits in which the REE are inferred to be weakly adsorbed onto clay minerals. Similar deposits elsewhere might provide alternative supply for these high-tech metals, but the adsorption mechanisms remain unclear and the adsorbed state of REE to clays has never been demonstrated in situ. This study compares the mineralogy and speciation of REE in economic weathering profiles from China to prospective regoliths developed on peralkaline rocks from Madagascar. We use synchrotron X-ray absorption spectroscopy to study the distribution and local bonding environment of Y and Nd, as proxies for heavy and light REE, in the deposits. Our results show that REE are truly adsorbed as easily leachable 8- to 9-coordinated outer-sphere hydrated complexes, dominantly onto kaolinite. Hence, at the atomic level, the Malagasy clays are genuine mineralogical analogues to those currently exploited in China.

[1] School of Earth and Environmental Sciences, University of St. Andrews, St. Andrews KY16 9AL, UK. [2] School of Environment and Technology, University of Brighton, Brighton BN2 4GJ, UK. [3] GET, CNRS, IRD, UPS, University of Toulouse, Toulouse, France. [4] British Geological Survey, The Lyell Centre, Research Avenue South, Edinburgh EH14 4AP, UK. [5] College of Earth Sciences, Guilin University of Technology, 541006 Guilin, China. [6] School of Earth and Space Sciences, Peking University, 100871 Beijing, China. [7] Department of Geology and Pedology, Mendel University, Zemedelska 1, 61300 Brno, Czech Republic. [8] BIC Brno Spol. s.r.o., Technology Innovation Transfer Chamber, Purkyňova 648/125, 61200 Brno, Czech Republic. [9] Diamond Light Source, Physical Science, Harwell Science Campus, Didcot OX11 0DE, UK. ✉email: anoukborst@gmail.com

Regolith-hosted ion-adsorption deposits (IADs), formed by subtropical weathering of igneous rocks, are the world's primary source for heavy rare earth elements (HREEs, Gd-Lu and Y)[1,2]. These elements, along with some of the other rare earth elements (REEs), are considered critical to society because of their ubiquitous use in modern technologies and renewable energy solutions, and because limits on supply would hinder global economic and technological development[3–5]. Regolith-hosted IADs, also referred to as ion-adsorption clays, weathering crust elution- or laterite-hosted rare earth deposits[6–9], readily liberate the metals following mild acidification during addition of ammonium sulfate leach solutions[10]. In such deposits, the REEs are inferred to be weakly adsorbed onto clay minerals (dominantly kaolinite and halloysite), as well as oxides, at a range of structural sites, including broken edge sites, charged aluminol or siloxane groups at defects and isomorphic substitutions (e.g. $Al^{3+}$ for $Si^{4+}$, $Fe^{3+}$ for $Al^{3+}$), the hydration shells of exchangeable cations or by direct substitution of exchangeable cations [11,12]. However, such behaviour is also consistent with the dissolution of nanoparticulate secondary REE-bearing phosphates, (fluor)carbonates, or colloid particles[13]. Based on the assumption that the exchangeable REEs are clay adsorbed, the term IAD is ubiquitously used in the literature to describe easily leachable REE deposits associated with lateritic weathering. However, despite their global economic and scientific importance, the adsorption mechanisms of REEs in lateritic deposits of commercial value remain poorly understood and the coordination of REEs in such deposits has never been measured in situ.

The majority of economically exploited IADs occur in Southern China, where the REEs are hosted in the weathered crusts of igneous, mostly granitic, bedrock[7,8,12]. They account for c. 35% of China's total REE production and roughly 80% of global HREE supplies[4,6], despite being low grade (0.05–0.2 wt% total $RE_2O_3$, incl. $Y_2O_3$) and relatively low tonnage compared to hard-rock REE deposits associated with carbonatites and alkaline igneous rocks[1,14]. Economic exploitation of IADs is viable through REE extraction by low-cost in situ or heap leaching[8,12,15], a process that has had significant environmental impact in China[6]. The key requirement is that the majority of REEs are readily liberated using ionic solutions and are hence ion exchangeable[10].

Weathering profiles developed on protoliths with a primary enrichment in the mid and HREEs, such as peralkaline igneous rocks, have become a key target for exploration[16,17]. Many prospective weathering profiles, some of which are HREE enriched, have recently been studied outside China, including in Malawi, Madagascar, USA, Brazil, the Philippines, Laos, Thailand and Myanmar[18–25] (Fig. 1). These may provide alternative supply of HREE, but it is unclear whether the clays in these profiles are direct structural analogues to the economically exploited Chinese clay deposits and whether similar REE adsorption mechanisms operate within them[12].

This work compares economically mineralised regoliths from Southern China, developed on the Zhaibei granite, Yiangxi Province[17,26], to prospective regolith profiles on peralkaline granites and syenites of the Ambohimirahavavy complex, northern Madagascar[27–29] (Fig. 1). Using X-ray absorption spectroscopy (XAS), including X-ray absorption near-edge structure (XANES) and extended X-ray absorption fine structure (EXAFS), micro synchrotron X-ray fluorescence (μSXRF) element mapping, and scanning electron microscopy (SEM), we identify the microscale distribution and local bonding environment of REEs associated with clays and secondary minerals in the weathering profiles. Previous studies utilising XAS to investigate rare earths in Malagasy regoliths primarily presented Ce $L_3$-edge XANES as an indicator of redox behaviour[30,31], but the redox sensitivity of Ce means that it is decoupled from the rest of the

REEs. XAS was also used to study the speciation of Sc in laterite profiles of ultra-mafic rocks, where Sc was found to be adsorbed dominantly to iron oxides[32]. Although these studies provide important insights into the geochemical behaviour and deportment of Ce and Sc during weathering, they do not constrain the distribution and speciation of the economically significant elements HREE and Nd in association with lateritic clays developed on rare earth bearing igneous protoliths. Here, we measure Y K-edge and Nd $L_3$-edge X-ray absorption spectra, as proxies for HREE and light REE, respectively, from regolith samples and leachates to quantify the structural state of REEs associated with clay minerals as well as relict phases. Our results provide novel constraints on the speciation and coordination of these elements in regoliths of economic interest, and demonstrate that the rare earths are genuinely adsorbed to clay minerals.

## Results

**Locality descriptions**. Regolith profiles in the Zhaibei region, southern Jiangxi province, developed through subtropical weathering of Jurassic peraluminous biotite and muscovite granites (188 ± 0.6 Ma)[17]. The laterites range in thickness between 5 and 30 m[26]. The Zhaibei granite is dominantly composed of quartz, K-feldspar, plagioclase and biotite, with small quantities of muscovite and amphibole. The REEs are dominantly hosted in the micas and amphibole, as well as accessory phases, such as zircon, monazite, Ca-REE fluorcarbonates, fergusonite-(Y), aeschynite-(Y), xenotime, titanite, rutile, ilmenite and fluorapatite[17,26]. Samples were obtained from surface-exposed profiles in a 12-m-thick mineralised laterite (LJ316, Fig. 2). The exchangeable fraction of REEs in the LJ316 profile demonstrates a moderate LREE enrichment and negative Ce and Eu anomalies (Supplementary Fig. 1), containing up to 29–44% HREEs (Supplementary Fig. 2).

The Malagasy regolith profiles developed on highly heterogeneous bedrock comprising silica-oversaturated and -undersaturated syenites, peralkaline granites and pegmatites, as well as volcanic lithologies belonging to the Cenozoic Ambohimirahavavy subvolcanic ring complex (c. 23 Ma)[27–29,33,34]. The country rocks (Mesozoic marl, limestone and mudstone) and the main syenites have low abundances of primary REE-bearing minerals, whereas the peralkaline granite, nepheline syenite and pegmatite dykes contain a wide range of REE-rich minerals, including REE fluorcarbonates, zirconosilicates, silicates, oxides and minor phosphates. Many of these comprise a secondary paragenesis formed through late-magmatic hydrothermal alteration of primary igneous phases, such as eudialyte-group minerals (EGMs) (Na-Ca-zirconosilicates)[35], alongside Nb minerals, such as members of the pyrochlore group and aeschynite-(Y)[28,34]. The Malagasy regolith profile is up to 30 m thick and comprises, from the bottom up: unaltered bedrock, saprock, saprolite and a pedolith made up of a mottled zone, a ferruginous zone and a topsoil (Fig. 2). The supergene mineralogy includes gibbsite, Mn and Fe oxyhydroxides and clay minerals, primarily kaolinite, minor halloysite (both 7 and 10 Å) and illite[28].

**Sample characterisation**. The Zhaibei pedolith sample chosen for XAS analyses was taken from the upper 2 m of the LJ316 laterite profile, which had the highest exchangeable REE fraction (1000.9 mg kg$^{-1}$, 37% HREEs; Supplementary Figs. 1 and 2; Supplementary Table 1). SEM analyses show the mineralogy to be dominantly kaolinite, K-feldspar, quartz and iron oxides, with minor illite, micron-sized zircon and rutile-bearing pseudomorphs after titanite (Fig. 3a, b). X-ray diffraction (XRD) of the sub-2-μm fraction as oriented powders indicates that kaolinite ($Al_2Si_2O_5(OH)_4$) is the dominant clay mineral, with minor

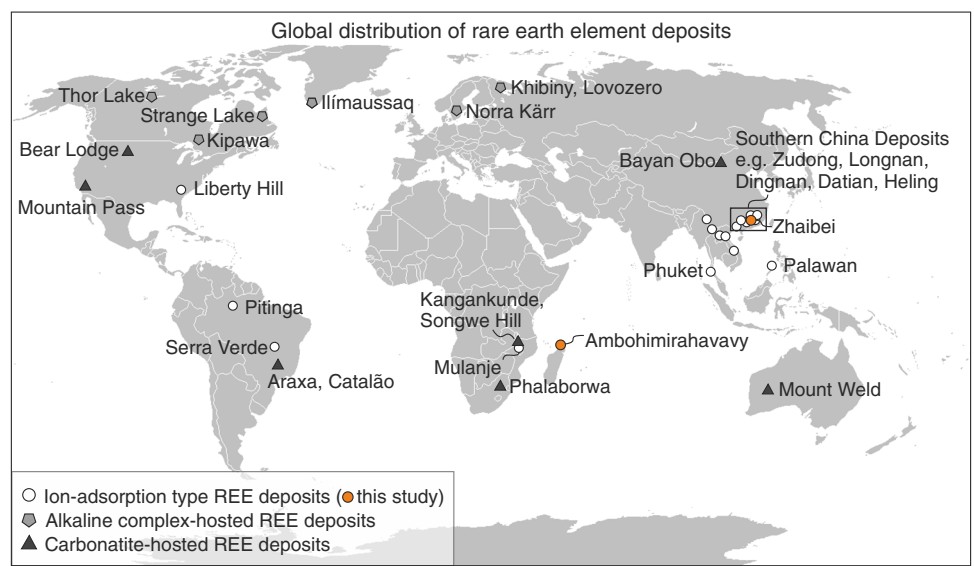

**Fig. 1 Global distribution of rare earth element deposits.** Regolith-hosted ion-adsorption-type REE deposits (shown in white circles) dominantly occur in the tropics and subtropics and currently provide most of the world's heavy REE supply. The majority of economically exploited regolith-hosted REE deposits occur in China, where REEs are associated with clay minerals in granitic weathering profiles. The two localities studied here are marked in orange: the Ambohimirahavavy complex in Madagascar and the Zhaibei granite in China. Other economically significant rare earth prospects are dominantly hard-rock deposits associated with alkaline igneous rocks and carbonatites, some are shown as grey pentagons and black triangles, respectively.

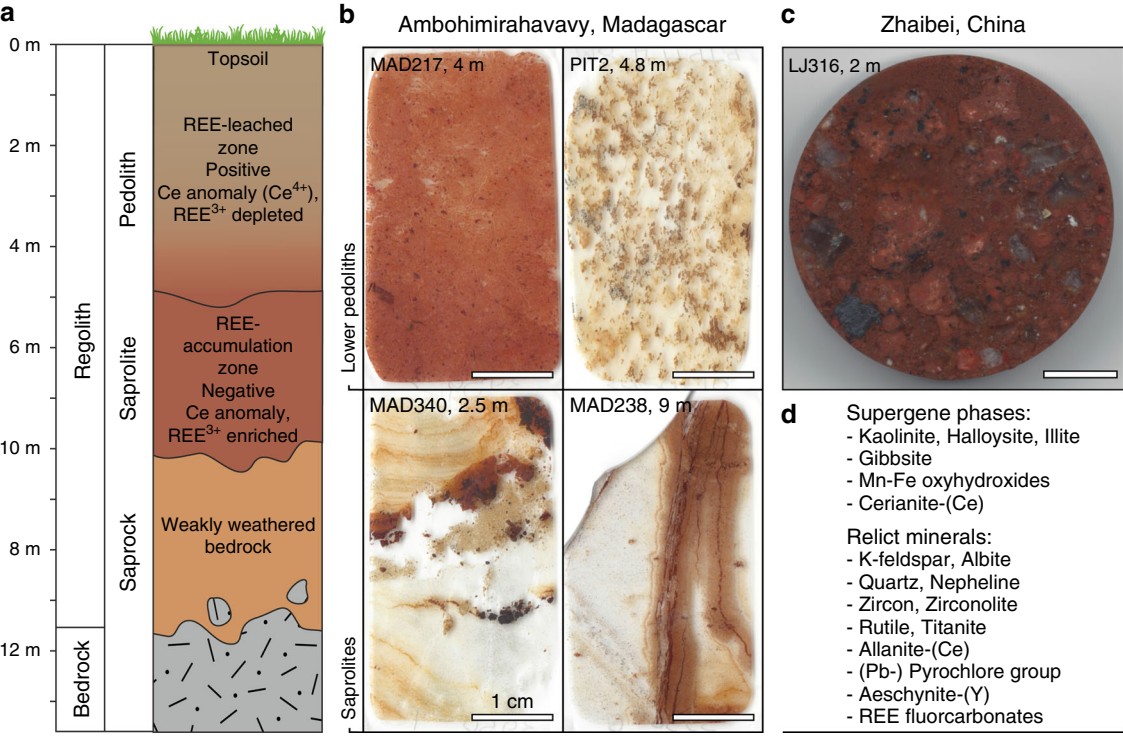

**Fig. 2 Schematic regolith profile and studied samples. a** Schematic regolith profile indicating leaching in the top of the weathering profile (pedolith) and accumulation of rare earth elements (REEs) in the saprolite, in which REEs are inferred to be adsorbed to clay minerals. The presence of $Ce^{4+}$ in the leached and oxidised top zone of the pedolith, hosted in insoluble cerianite-(Ce), commonly leads to positive Ce anomalies, whereas complementary negative Ce anomalies are recorded in the REE-accumulation zone (saprolite). The saprolite is underlain by weakly weathered bedrock (saprock) and unaltered bedrock. **b** Thin section photographs of the rare earth-rich lower pedolith and saprolite samples from Madagascar. White scale bars are 1 cm. **c** Photograph of the resin mount from the pedolith sample from China. Scale bar is 1 cm. **d** List of supergene and relict mineral phases identified in the regolith samples.

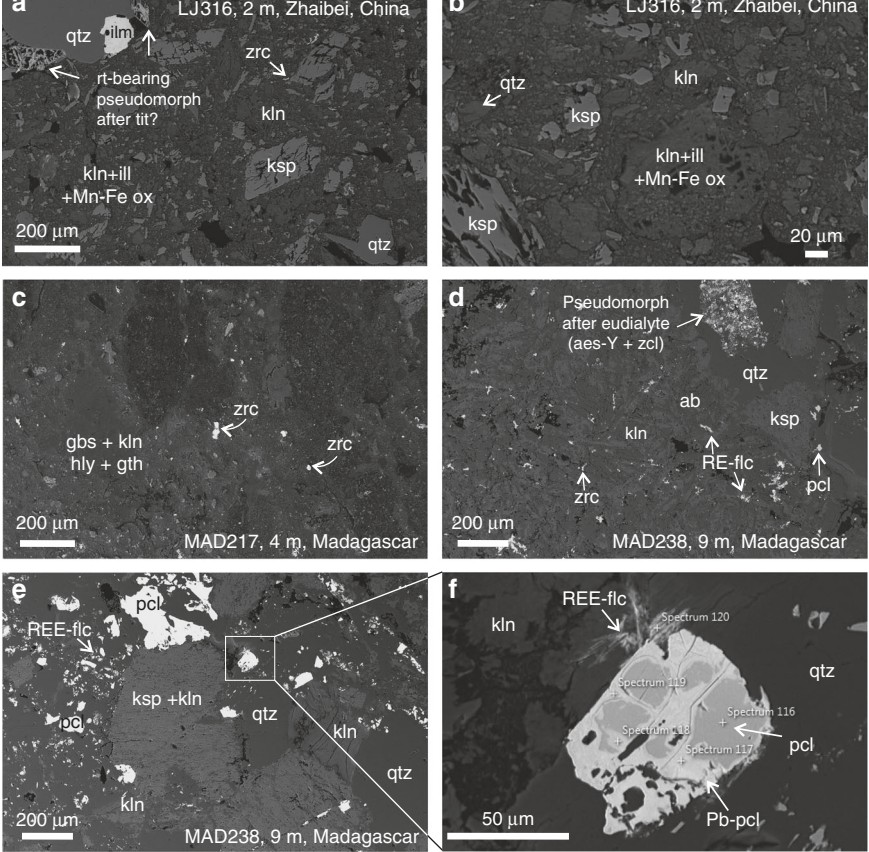

**Fig. 3 Petrography of the studied samples. a, b** Backscatter electron images of relict grains of alkali feldspar (ksp) and quartz (qtz) in a fine-grained matrix of kaolinite (kln), minor illite (ill) and Mn and Fe oxyhydroxides (Mn-Fe ox) in the Zhaibei pedolith sample. Refractory magmatic phases containing appreciable high field strength elements (HFSEs), including Zr, Ti, Nb and rare earth elements (REEs), include zircon (zrc), ilmenite (ilm) and lozenge-shaped pseudomorphs containing rutile (rt), likely after magmatic titanite (tit). **c–f** Malagasy samples contain alkali feldspar (ksp, ab), quartz, gibbsite (gbs), kaolinite, minor halloysite (hly) and goethite (gth). Dominant HFSE-bearing phases include zircon (zrc), zirconolite (zcl) and aeschynite-(Y) (aes-Y), which occur in pseudomorphs after eudialyte, as well as pyrochlore (pcl), which shows late-magmatic alteration to Pb-rich pyrochlore (Pb-pcl) and REE fluorcarbonates.

gibbsite and no discernible halloysite-10 Å (Supplementary Fig. 3). Since halloysite-10 Å (Al$_2$Si$_2$O$_5$(OH)$_4$·2H$_2$O) can readily dehydrate to halloysite-7 Å (Al$_2$Si$_2$O$_5$(OH)$_4$), some halloysite-7 Å may be present, but this is difficult to quantify or distinguish from kaolinite using XRD[36].

Selected samples from the Ambohimirahavavy complex include pedoliths and saprolites from a hand-dug pit in a regolith profile inferred to overlie a Si-undersaturated syenite, and drill cores of regolith profiles on REE mineralised peralkaline granite and pegmatite dykes (Supplementary Data 1). Samples analysed by XAS were chosen to reflect a range in exchangeable REE contents (Supplementary Figs. 1 and 2, Supplementary Table 1), and to compare REE structural states in regolith material from different protoliths and variable depths (Fig. 2).

PIT2 is a hand-dug pit on a regolith profile overlying an evolved Si-undersaturated microsyenite[28]. The selected sample for XAS analyses comes from a depth of 4.8 m, where the PIT2 profile shows the highest grade in exchangeable REEs (1963 p.p. m. REEs, 15% HREEs; Supplementary Fig. 2, Supplementary Table 1). SEM analyses indicate the presence of kaolinite, K-feldspar, Fe and Mn oxyhydroxides with accessory zircon and traces of cerianite-(Ce) locally associated with Fe-Mn-rich areas (Fig. 4c–f). These samples were investigated for their mineralogy by Estrade et al.[28] who reported 5–50 mass % of kaolinite, with subsidiary, but significant halloysite and minor gibbsite. Samples MAD217 and MAD238 are a pedolith and a saprolite taken from

4 and 9 m depth, respectively, in a drill core directly over REE mineralised granitic pegmatite dykes[28]. The former has an exchangeable REE fraction of 783.43 p.p.m. (17% HREEs) and is dominantly composed of kaolinite, quartz and iron oxides with microscopic grains of refractory zircon (Fig. 3c). Saprolite sample MAD238 was taken where REE mineralised peralkaline dykes cut mudstone[28]. The dyke material is composed of albite, K-feldspar and quartz with minor amounts of zircon, Ca-REE fluorocarbonates, pyrochlore-group minerals and aeschynite-(Y), occurring interstitially between clays and within pseudomorphs after a primary Na-Ca-HFSE phase, inferred to be an EGM (Fig. 3d). Some REE fluorocarbonates are associated with partially decomposed pyrochlore-group minerals with late-magmatic or hydrothermal alteration zones enriched in Pb (Fig. 3e, f). Although clearly REE mineralised, MAD238 has the lowest concentration of exchangeable REEs (484 p.p.m., 27% HREEs; Supplementary Fig. 2, Supplementary Table 1), consistent with most REEs being hosted in refractory REE phases. Sample MAD340 (634 p.p.m., 32% HREEs) is a saprolite sample collected from a surface road cut within the syenite ring dyke of the complex. It is dominantly composed of kaolinite with accessory zircon and rutile (Fig. 4g).

**Element mapping.** µSXRF mapping was used to visualise the microscale distribution of Y associated with clays, Fe-Mn oxides and other supergene or relict mineral phases at both sites.

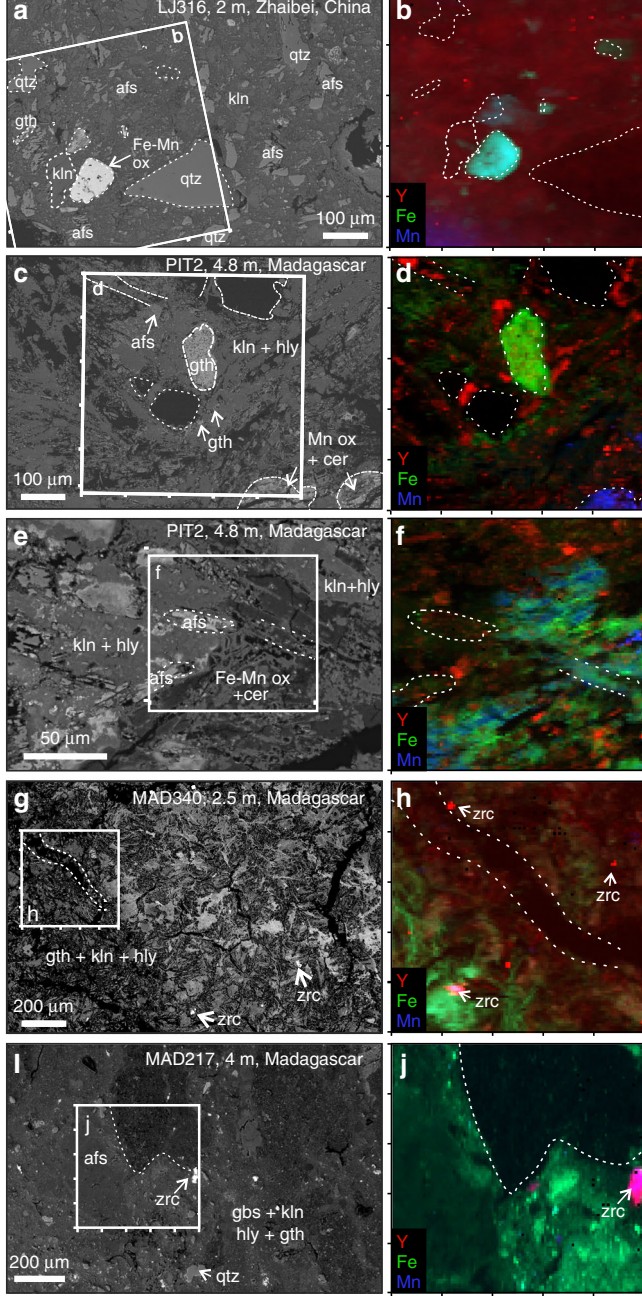

**Fig. 4 Mineralogy and synchrotron element maps.** Backscatter electron images of studied regolith samples with accompanying synchrotron micro X-ray fluorescence maps to show the distribution of Y (red), Fe (green) and Mn (blue) in different mineral phases. **a** Backscatter image of the Zhaibei pedolith sample containing relict quartz (qtz) and alkali feldspar in a matrix of kaolinite (kln) and grains of goethite and Fe-Mn oxyhydroxides (Fe-Mn ox). White box indicates location of Y-Fe-Mn element map shown in **b** showing heterogeneous enrichment of Y (red) with localised hotspots in the kaolinite matrix. **c-f** Malagasy pedolith sample containing goethite (gth) and Mn oxyhydroxides with traces of cerianite-Ce (cer) in a matrix of kaolinite (kln) and halloysite (hly). White boxes in **c**, **e** correspond to Y-Fe-Mn element maps in **d**, **f**, respectively, both showing strong local enrichments of Y along the margins of clay minerals. **g** Strongly weathered saprolite sample from Madagascar showing **h** medium clay-hosted Y enrichment and localised Y hotspots associated with microscopic grains of relict zircon (zrc). **i** Fine-grained gibbsite (gbs) and goethite-rich pedolith sample with **j** low concentrations of clay-hosted Y and local Y enrichments associated with relict zircon.

Different REE-bearing phases identified in the element maps were targeted for Y K-edge and Nd $L_3$-edge XANES and EXAFS analyses. Representative element maps are shown in Fig. 4. μSXRF maps of the Zhaibei pedolith demonstrate localised enrichments of Y in and around clay minerals, as well as Fe-Mn-rich areas containing haematite, goethite and Mn oxyhydroxides (Fig. 4a, b). The hotspots of Y associated with clay minerals occur where no distinct bright REE-rich phases are identified in back-scatter and by SEM-based energy-dispersive X-ray (EDX) analyses, and may represent nanoscopic REE phases or heterogeneously adsorbed Y. Element maps for the Malagasy samples with the highest leachable HREE fractions show similar elemental distributions, with Y visibly enriched along the margins of clay aggregates, possibly correlating with areas of finer crystal sizes and higher porosity. Within saprolites MAD238 and MAD340, a significant portion of the Y is associated with relict bedrock phases of zircon, pyrochlore, zirconolite or aeschynite-(Y) as identified by SEM analyses (Figs. 3d–f, 4g, h). In lower pedolith sample MAD217, most Y is hosted in <30-μm-sized grains of zircon (Fig. 3c), while the clays show no particular enrichments in Y (Fig. 4i, j). This is consistent with the relatively low leachable HREE fraction (17%) in this sample. X-ray absorption spectra were collected from all identified phases with variable Y concentrations to determine whether localised Y enrichments represent nanoparticles of primary or secondary REE-rich phases, such as zircon, pyrochlore-group minerals or Ca-REE fluorcarbonates within the clays, or are adsorbed to clay surfaces.

**Yttrium X-ray absorption spectra.** Normalised Y K-edge XANES spectra from the samples and standards are shown in Fig. 5. The standards were measured to demonstrate a range of REE coordination states and point symmetries, allowing for a qualitative comparison with the Y K-edge XANES obtained from the regolith samples. Mineral standards in which Y occupies low point-symmetry sites, that is, parisite-(Ce), bastnäsite-(Ce) (CN = 9) and vitusite-(Ce) (CN = 6), as well as $Y^{3+}$ in aqueous solution (CN = 8), demonstrate relatively featureless Y XANES spectra with sharp white lines at 17056 eV and a broad peak ~17,104 eV (Feature C, Fig. 5a). XANES result from atom-to-atom resonances that are promoted by elements occupying sites with high point symmetry. Hence, higher symmetry minerals in which REEs occupy higher point-symmetry sites show more pronounced XANES features. For example, Y-doped $NdPO_4$ (monazite structure, CN = 9), zircon and xenotime (both natural and synthetic Nd-doped $YPO_4$, CN = 8) have a characteristic bump at 17,064 eV on the shoulder of the main XANES peak (Feature B, Fig. 5a). This feature is visible also in rinkite and nacareniobsite-(Ce) (Nb-Ti disilicates) in which REEs occupy seven-coordinated $M^H$ and $A^P$ sites[37,38]. Spectra for $Y_2O_3$ (CN = 6) show a second peak at 17,064 eV (Feature B) that is as prominent as the main peak at 17,056 eV, comparing well to published Y K-edge XANES spectra for cubic $Y_2O_3$[39]. EGMs are characterised by a double-peaked XANES with a minor peak at 17,053 eV (Feature A) and a broad maximum between A and B at 17,061 eV[40]. The spectra show a systematic shift in the position and shape of the broad peak around 17,104 eV (Feature C), shifting to higher energy with decreasing coordination numbers (CNs) (from 17,104 eV for Y in low symmetry 11- to 8-fold sites, to 17,110 eV for phases with Y in six-coordinated sites[40]).

Figure 5b, c shows Y XANES for relict minerals and supergene phases in the laterite samples. In both the Madagascar and Chinese samples, Y XANES measured on kaolinite with medium to high Y concentrations are identical (Fig. 5b, c). They compare most closely to Y XANES of parisite-(Ce) and Y in standard

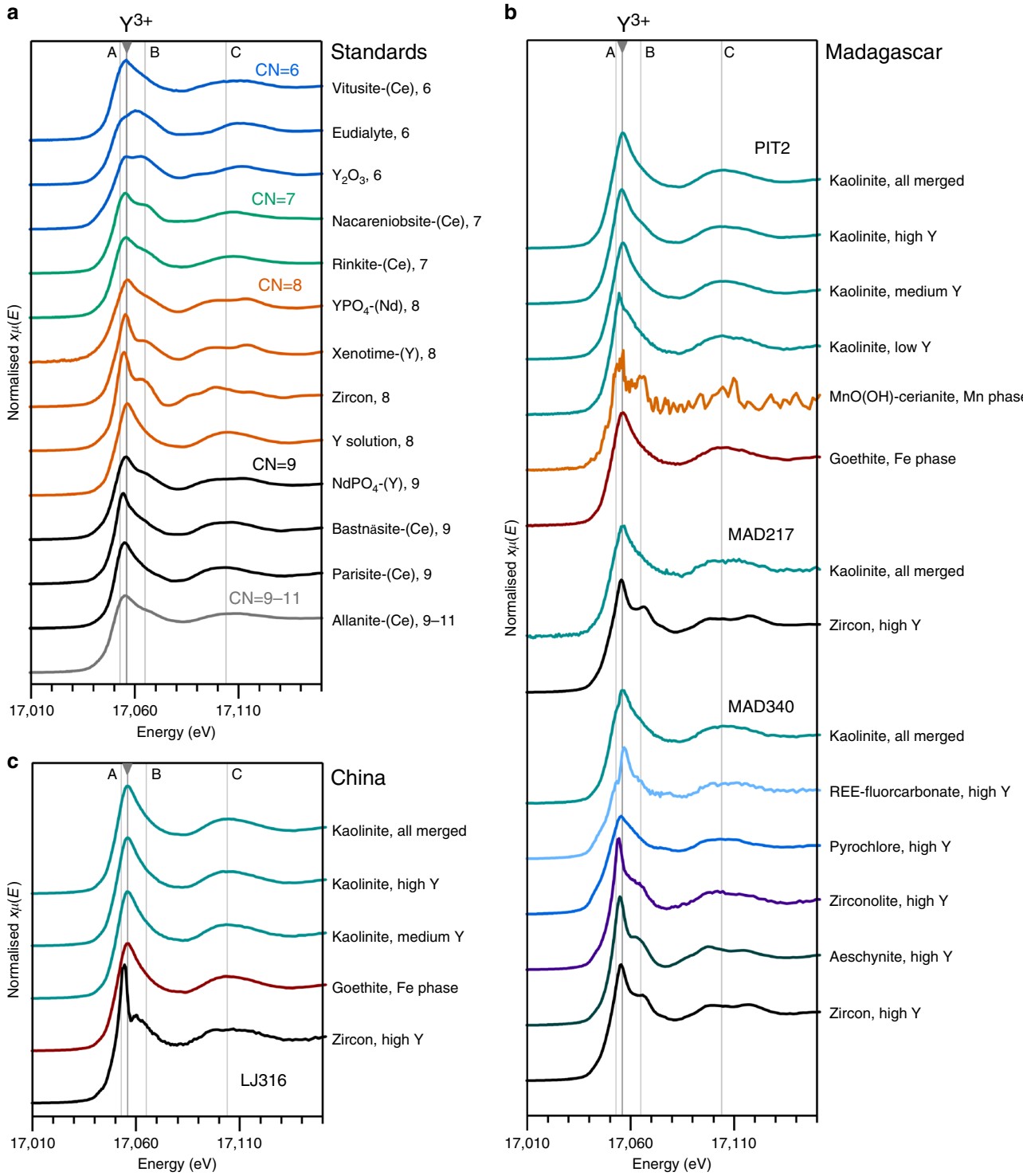

**Fig. 5 Yttrium X-ray absorption spectra. a** Yttrium K-edge X-ray absorption near-edge structure (XANES) spectra for rare earth element-bearing mineral standards and solution, vertically arranged by decreasing coordination number of the rare earth site (CN). **b** Y XANES for supergene and relict mineral phases in Madagascar samples. **c** Y XANES for supergene and relict minerals in the Chinese sample. In all panels, the main adsorption features are indicated by grey lines at 17,056 eV ($Y^{3+}$, main peak), 17,053 eV (Feature A), 17,064 eV (Feature B), and 17,104 eV (Feature C). Details on mineral standards are provided in Supplementary Data 1 and X-ray absorption data provided in Supplementary Data 2.

solution, in which Y occupies 9- and 8-fold coordinated sites, respectively. Specifically, the position of feature C in the medium to high Y kaolinite XANES lies at 17,104 eV, comparable to the low point-symmetry standards with Y in 8- to 9-fold sites (Fig. 5a).

Notably different are the XANES spectra measured on kaolinite areas with background to low concentrations of Y (PIT2, Fig. 5b) which demonstrate a peak at lower energy and a shoulder at Feature B, similar to the XANES of laterite-hosted zircon and zirconolite spectra (below). Yttrium XANES from goethite grains

in sample LJ316 and PIT2 (Fig. 4b, d) are comparable to the clay-hosted Y XANES, suggesting a similar structural state for REEs adsorbed to goethite (Fig. 5b, c) and clay at low concentrations. A XANES spectrum from the Mn-rich area (blue area in Fig. 4d), in which SEM suggests the presence of MnO(OH) with traces of cerianite-(Ce), yields noisy Y XANES reflecting low Y concentrations (Fig. 5c).

Yttrium hotspots in the μSXRF maps of the Zhaibei regolith (Fig. 4b) are confirmed by SEM analyses to correspond to microscopic grains of zircon, as the only significant relict REE-bearing phase. These grains yield Y XANES spectra that, although noisy, are markedly different from the clay-hosted Y XANES. Refractory REE phases identified by SEM in the Madagascar saprolites and pedoliths (i.e. the Y hotspots in μSXRF maps) include zircon, pyrochlore-group minerals, aeschynite-(Y) and zirconolite as confirmed by SEM (Fig. 5c). Merged XANES spectra from relict grains of zircon in MAD340 and MAD217 compare well to the Y XANES of 8-fold coordinated high symmetry Y standards (zircon and xenotime), which show pronounced features at 17,056 eV (Feature B) and a double peak ~17,104 eV (Feature C), indicating Y is hosted in the zircon lattice. It is of note that μSXRF mapping of Malagasy samples by Ram et al.[31] demonstrated a positive correlation of REE + Y with Zr associated with clay minerals in samples containing no relict zircon, indicating local co-sorption of $Zr^{4+}$ with $REE^{3+}$ in the most strongly weathered samples. No XAFS data are currently available for Zr in these profiles to confirm sorption of Zr to clay surfaces. Aeschynite-(Y), a Y-Ti-Nb oxide in which REEs occupy 8-fold sites[41], and zirconolite ($CaZrTi_2O_7$) in which REE substitute for 8-fold Ca[42], show similar XANES spectra to zircon and xenotime standards. Both phases occur within fine-grained pseudomorphs after EGMs in the MAD238 saprolite (Fig. 3d). XANES spectra from relict pyrochlore-group minerals in the same sample (both Pb-poor and Pb-rich zones, Fig. 3e) demonstrate a lower white line and less pronounced XANES features. Yttrium enrichments in μSXRF maps near the margins of the pyrochlore grains are interpreted as micron-sized secondary REE fluorcarbonates (Fig. 3e) formed by the breakdown of primary pyrochlore, and yield a somewhat noisy but broad Y XANES that is comparable to the fluorcarbonate standards.

**Neodymium X-ray absorption spectra.** Normalised Nd $L_3$-edge XANES spectra for the samples and standards are shown in Fig. 6. All minerals show sharp white lines at 6215 eV with narrow peaks and minor variations in the relative heights and positions of Features A and B (grey lines in Fig. 6). Despite tight windowing of the secondary X-rays, absorption features at the Ce $L_2$-edge are visible in the Nd pre-edge region for all natural standards and samples. Relative variations in height of the $Ce^{3+}$ and $Ce^{4+}$ absorption peaks reflect natural variations in Ce oxidation state associated with different phases[30]. Cerium peaks observed in the pre-edges of the clay-hosted Nd XANES from both Madagascar and China demonstrate the presence of Ce, which is dominated by $Ce^{3+}$ with variable $Ce^{4+}$. Peaks for $Ce^{4+}$ (at 6179 eV) are most prominent in zircon and in MnO(OH)-rich areas from the Malagasy laterites, consistent with the presence of cerianite ($CeO_2$) as determined by SEM (Fig. 4c, d) and the formation of secondary zircon under oxidising conditions. The deportment of Ce with Fe/Mn oxides was previously noted in the Malagasy profiles[28,30,31] and is a common feature in regolith profiles globally[12,18], reflecting the effective oxidation of $Ce^{3+}$ to $Ce^{4+}$ by Fe and Mn oxyhydroxides, leading to characteristic positive Ce anomalies in the oxidised REE-leaching zones and negative Ce anomalies in the REE-accumulation zones[12,30,43,44]. We note that

Ce adsorption features are higher in the XANES spectra of supergene phases with lower Y concentrations (Fig. 6b), demonstrating heterogeneous distribution of Y/Ce in the Malagasy PIT2 sample. The relative height of the Ce adsorption peaks, as well as the $Ce^{3+}/Ce^{4+}$ ratios, with respect to Nd and Y are more consistent in the Zhaibei sample.

The clay- and oxide-hosted Nd XANES are identical within individual laterite samples, but the Chinese spectra slightly vary from those of Madagascar, with Feature B (6286 eV) being more prominent in spectra from the Malagasy PIT2 sample than in those from the Zhaibei sample. With exception of Feature B in PIT2, the clay-hosted Nd XANES most closely resemble spectra of the fluorcarbonates, $NdPO_4$, steenstrupine and allanite. Noisier Nd spectra for the goethite and Mn oxyhydroxide/cerianite-(Ce) reflect relatively low concentrations of Nd associated with those phases. Standards in which LREEs are inferred to occupy higher symmetry sites (yttrofluorite, xenotime (YPO$_4$-Nd), $Nd_2O_3$) show more pronounced Nd XANES features (several or higher peaks at Feature A, or a stronger Feature B, Fig. 6a) and higher white lines than XANES of minerals in which LREEs occupy lower symmetry, or multiple sites[40]. XANES for Nd in aqueous solution (in weak $HNO_3$) demonstrates the highest white line among the standards, and additional features that are not observed in the other spectra (Fig. 6a). From this we infer that Nd in aqueous solution occurs as highly ordered and symmetric hydrated Nd-OH complexes. Unlike the Y K-edge XANES, no systematic variations are observed in the height or shape of Nd XANES peaks with increasing CN in the standards. Hence, we infer that the Nd $L_3$-XANES is dominated more by the point symmetry of the site and, compared to Y, is relatively insensitive to the CN.

**Coordination of yttrium.** The local atomic structure of Y associated with clay minerals in the Chinese and Malagasy laterites is further constrained by fitting the Y K-edge EXAFS spectra[45]. EXAFS signals for the clay-hosted Y were merged to improve signal-to-noise ratios and fitted to a theoretical scattering path of nearest-neighbour oxygens to constrain the local structure around the absorbing Y atom (e.g. CN and bond distances). From samples MAD217 and MAD238 only three and two spectra, respectively, were useable for quantitative analyses and were therefore merged together. The $k^2$-weighted EXAFS oscillations and phase-corrected radial distribution functions of the merged spectra and their fits are shown in Fig. 7. All radial distribution functions (Fig. 7b) demonstrate a single peak, representing the first coordination sphere of scattering atoms surrounding Y at an average distance of c. 2.4 Å. These can be attributed to the oxygens of water molecules, or the hydroxides and oxygens within the structural framework of the clay mineral (Fig. 7b). No further peaks are visible beyond the first shell of oxygens, which might indicate longer-range order, such as nearby Al or Si atoms within the structure of the clay or other crystalline solids. Refined parameters from the EXAFS curve-fitting analysis of the standards and samples are listed in Tables 1 and 2, respectively. Details of the fitting procedures are described in the "Methods". Least-squares fitting models of the Chinese and Malagasy clay EXAFS yield a local coordination with 7.9–8.3 ± 0.9 oxygens at a distance of $2.35 - 2.38 \pm 0.01$ Å around the central Y atom (Fig. 7b, Table 2). The EXAFS results of the clay-hosted $Y^{3+}$ compare well to that of $Y^{3+}$ in aqueous solution (Fig. 7a), which yields a best fit with 8.2 ± 0.4 oxygens at a radial distance of 2.37 ± 0.01 Å (Table 1).

**Discussion**
The clay fractions in the studied laterite samples are dominated by kaolinite, and in the case of Madagascar, minor halloysite

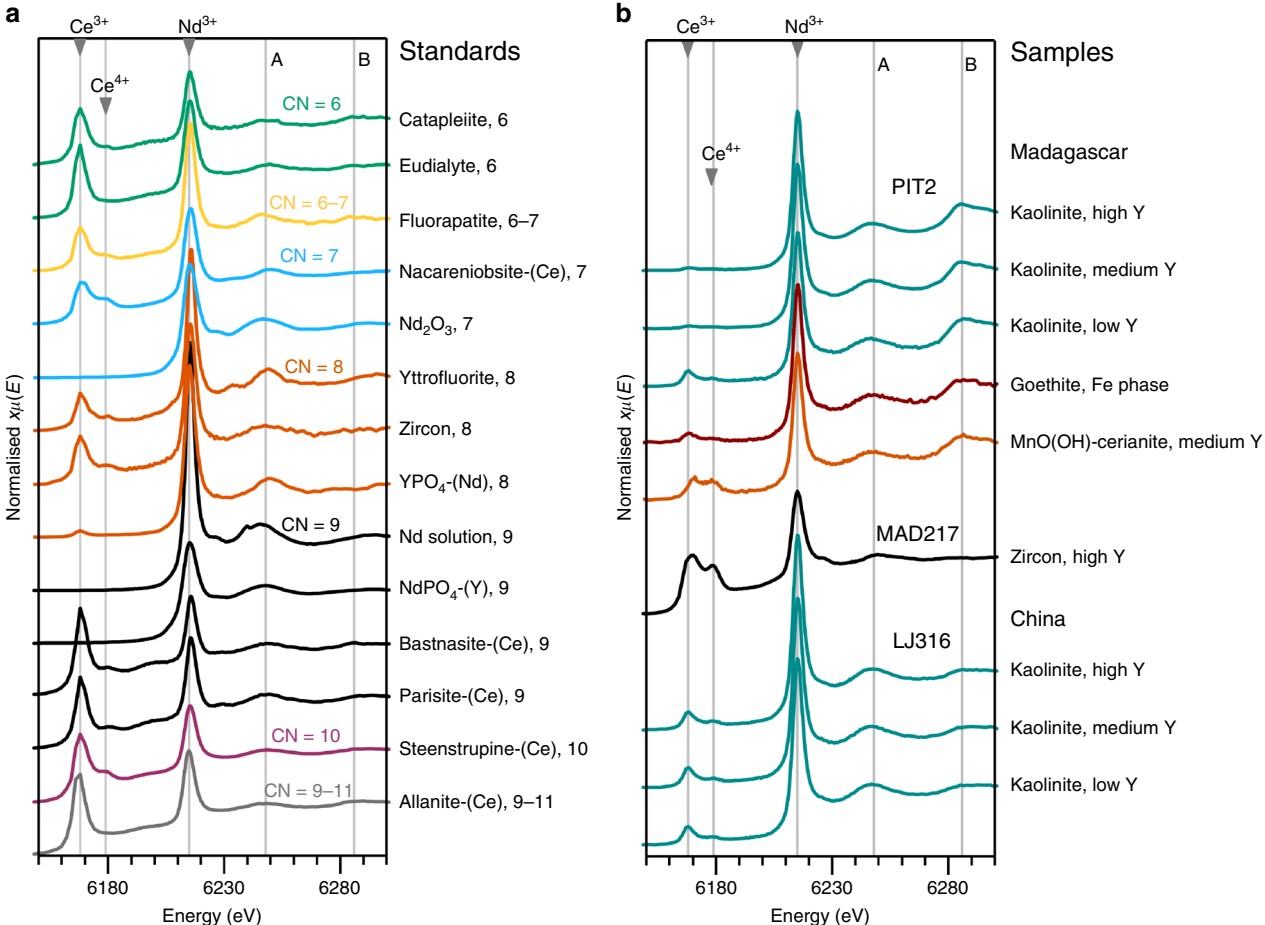

**Fig. 6 Neodymium X-ray absorption spectra. a** Neodymium $L_3$-edge X-ray absorption near-edge structure (XANES) for mineral standards and Nd in solution, vertically arranged by decreasing coordination number (CN). Cerium $L_2$-edges observed in the Nd pre-edge demonstrate variable Ce oxidation states ($Ce^{3+}$ at 6168 eV and $Ce^{4+}$ at 6179 eV) in the natural mineral standards. The main Nd absorption features are marked by grey lines at 6215 eV ($Nd^{3+}$, main peak), at 6248 eV (Feature A) and 6286 eV (Feature B). **b** Nd XANES spectra for clays and relict minerals in the Chinese and Malagasy samples. Kaolinite spectra from both sites demonstrate variable peaks for $Ce^{3+}$ and minor $Ce^{4+}$. Absorption of $Ce^{4+}$ (6179 eV) is most prominent in the zircon and the Mn oxyhydroxide, the latter containing traces of cerianite-(Ce). Grey lines are as in **a**.

(10 Å and possibly 7 Å)[28,31]. Other supergene phases include gibbsite, goethite, Mn oxyhydroxides and cerianite-(Ce). SXRF element mapping of the Zhaibei and Ambohimirahavavy pedolith samples with the highest concentrations of exchangeable REEs show that REEs are dominantly associated with clay minerals. In the saprolite samples from Madagascar, a significant portion of REEs are hosted in refractory mineral phases, such as zircon, pyrochlore, zirconolite, aeschynite-(Y) and REE fluorcarbonates. XANES of the clay-hosted Y and Nd from both Madagascar and China are similar to XANES spectra of REE fluorcarbonates, notably parisite-(Ce), although SEM examination shows these phases are not visible in the areas measured. The rare earth site in parisite-(Ce) is nine coordinated, surrounded by six oxygens and three fluorine atoms in a site of low point symmetry[46]. Yttrium XANES for both the Malagasy and Chinese clays also compare well to XANES obtained from Y in aqueous solution (Fig. 5). Speciation studies of REEs in aqueous fluids at ambient temperatures and neutral pH suggest that the smaller HREEs (Gd-Lu) favour an 8-fold hydration sphere (Fig. 7), whereas the larger light REEs (La-Eu) favour 9-fold coordination[47–49]. The similarity between clay-hosted Y and Nd XANES, those of fluorcarbonates and REEs in aqueous solution, suggest a similar 8- or 9-fold coordination sphere for the REEs associated with kaolinite and halloysite. EXAFS analysis further constrains the local atomic structure of Y hosted in clay minerals to be surrounded by 7.9 to

8.3 ± 0.9 oxygens at a radial distance of 2.35 – 2.38 ± 0.01 Å, and without further scattering of atoms up to a radial distance of 5 Å. This indicates a coordination sphere within error of that obtained for $Y^{3+}$ in aqueous solution (Table 1, Fig. 7).

To further interpret these EXAFS results, we briefly review the structure and metal sorption mechanisms of kaolinite-group minerals. Metal sorption on kaolinite and halloysite, as 1:1 layer clays, can take place either at charged edge sites or at neutral or charged basal surfaces[11,50]. Permanent charge resulting from isomorphic substitution for $Al^{3+}$ and $Si^{4+}$ in the octahedral and tetrahedral layers is relatively limited in the kaolinite group, as is interlayer ion exchange due to relatively strong bonding between the 1:1 layers[11]. This typically gives neutral siloxane (Si-O) surfaces at the tetrahedral layer[51], and precludes significant adsorption of hydrated ions or hydroxyl complexes on the siloxane surfaces[52,53]. Instead, most charge in kaolinite-group clays is pH dependent and occurs on the aluminol (Al-OH) surfaces of the octahedral layers (also the "gibbsite basal planes"), or on hydroxyl groups at edge sites and defects ("terminal OH groups")[12,36,54]. With limited isomorphic substitutions and interlayer ion exchange, the overall cation exchange capacity of kaolinite-group minerals is typically much lower than that of 2:1 layer clays, such as smectite and illite, which exhibit a much wider range of substitutions[50]. Nevertheless, kaolinite and halloysite are the dominant clay minerals in regolith-hosted REE deposits and

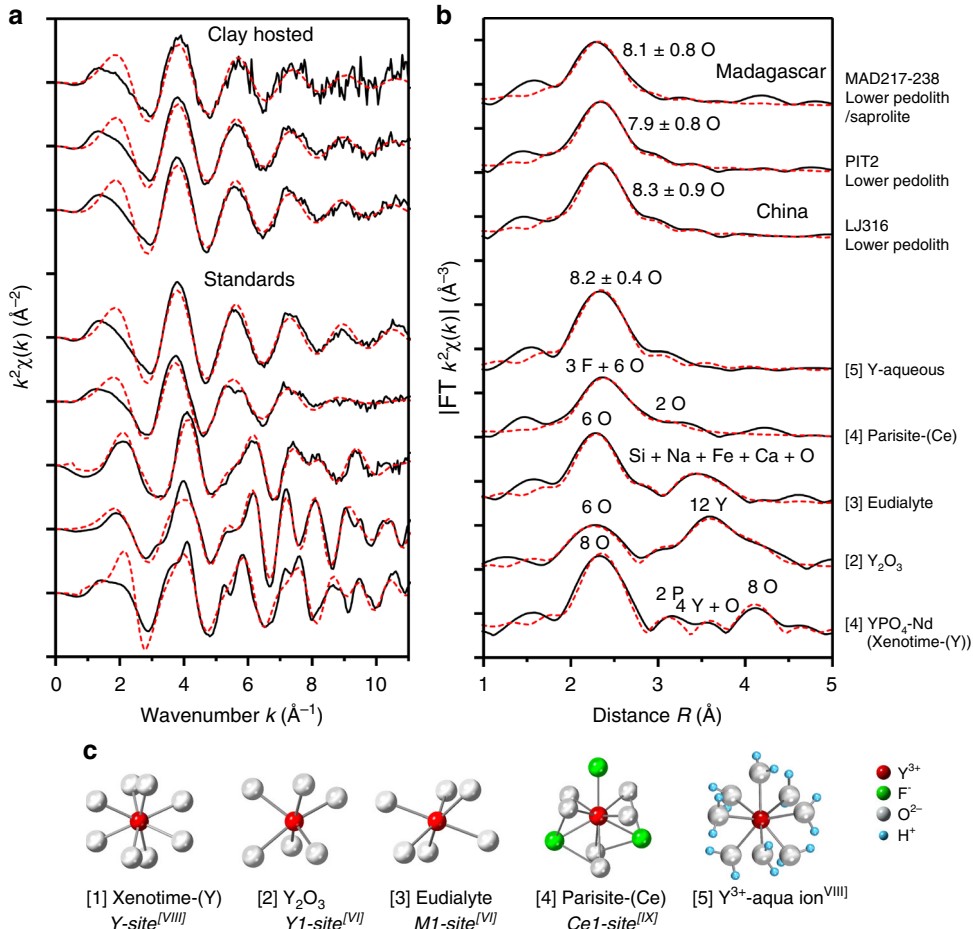

**Fig. 7 Coordination state of clay-hosted Y and selected standards. a** $k^2$-Weighted X-ray absorption extended fine structure (EXAFS) data of the Y K-edge in the Chinese and Malagasy clays and selected standards, showing EXAFS oscillations as a function of wavenumber $k$ (Å$^{-1}$). Experimental data are shown in solid black lines and least-square fits derived in Artemis in red dashed red lines. **b** Corresponding phase-shifted Fourier transform (radial distribution) functions derived from the EXAFS data. Each peak represents a shell of atoms surrounding the central Y atom at a certain radial distance $R$ (Å), with the height of the peak corresponding to the number and type of atoms (annotated) and the position of the peaks corresponding to their average distance to central Y atom. The clay-hosted Y spectra demonstrate a single shell at a radial distance of c. 2.35–2.38 Å. **c** Crystallographic models used in the EXAFS fitting procedures, demonstrating the XRD determined site geometry for Y in the measured standards. For simplicity only the first coordination sphere is shown, which corresponds to the first shell in the radial distribution functions of **b**. The clay-hosted EXAFS results are consistent with an 8-fold coordination of Y, similar to that measured for Y$^{3+}$ in aqueous solution. References for crystal structures of the standards are provided in Supplementary Data 1.

their respective roles in adsorbing and fractionating REE within lateritic weathering profiles have been highlighted in numerous recent studies, for example, refs. [12,16,55–58].

In addition to physicochemical and microstructural properties, such as particle size, morphology, crystallinity, composition and purity[36,50,55], the cation adsorption capacity of kaolinite and halloysite strongly depends on the pH and ionic strength of the weathering solutions in which they occur[52,56]. For example, these determine whether metal ion complexes are adsorbed onto the clay surfaces as outer-sphere complexes or as inner-sphere complexes[52,59]. Outer-sphere complexes retain a full hydration sphere and are loosely adsorbed to the basal surface via electrostatic attraction, whereas inner-sphere complexes are partially desolvated such that the metal bonds directly with Al-O or Si-O groups on the aluminol or siloxane surfaces, or edge sites of clay particles, as illustrated in Fig. 8. This distinction is important as outer-sphere complexes are more easily leachable than inner-sphere ones. Peacock and Sherman[60] and O'Day et al.[61] showed that divalent cations (Cu$^{2+}$ and Co$^{2+}$, respectively) provided EXAFS with features related to a second shell of Al (or Si) atoms,

consistent with inner-sphere adsorption via one or two bridging oxygens. A second shell of Al/Si atoms expected between 3 and 4 Å is not visible in our EXAFS results. Surface complexes on the aluminol sheet will therefore be of the form [≡Al-O-Ln(H$_2$O)$_{7–8}$]$^+$, representing a total coordination sphere of 8–9 (Fig. 8).

With increasing pH, inner-sphere complexation becomes more dominant due to hydrolysis of edge and basal hydroxyl groups in the clay structure[52,61]. Inner-sphere complexes for trivalent ions generally maintain a similar coordination sphere in the first shell, but are distinguishable from outer-sphere complexes using EXAFS data by showing evidence for a second shell of Al and/or Si atoms in the tetrahedral and octahedral sheets[59,61,62]. The close comparison of the clay-hosted EXAFS fitting results with that of Y in aqueous solution, combined with the absence of a second shell in our EXAFS data, indicates that the majority of Y retains a full hydration sphere and is held to the kaolinite (or halloysite) by van der Waals forces between water molecules and oxygens on the aluminol basal surface.

Moldoveanu and Papangelakis[63,64] suggested permanently chemically bound hydroxyl complexes (for aluminol rather than

**Table 1 EXAFS fitting results for the coordination state of Y in selected mineral standards and aqueous solution.**

| Standard | Origin | Shell | Path | CN | $R$ (Å) | $\sigma^2$ (Å$^{-2}$) | $\Delta E_0$ (eV) | $S_0^2$ | $\chi^2$ | R-factor | Ref. |
|---|---|---|---|---|---|---|---|---|---|---|---|
| Y-aqueous | Synthetic, Inorganic Ventures | 1st | Y-O | 8.2 (4) | 2.37 (1) | 0.005 | −1.7 | 1 | 656 | 0.016 | [1] |
| Parisite-(Ce) | Muzo Mine, Bogota, Colombia | 1st | Y-F | 3 | 2.29 (2) | 0.003 (2) | 0 | 1 | 44 | 0.016 | [1] |
| | | | Y-O$_1$ | 6 | 2.46 (1) | 0.006 (3) | | | | | |
| | | 2nd | Y-O$_2$ | 2 | 3.19 (4) | 0.009 (7) | | | | | |
| Y$_2$O$_3$ | Synthetic, Johnsen Matthey | 1st | Y-O | 6 | 2.27 (1) | 0.005 (1) | −6.4 | 1 | 2125 | 0.011 | [2] |
| | | 2nd | Y-Y$_1$ | 6 | 3.53 (0) | 0.003 (0) | | | | | |
| | | | Y-Y$_2$ | 6 | 4.01 (1) | 0.004 (1) | | | | | |
| YPO$_4$-Nd | Synthetic, ORNL | 1st | Y-O$_1$ | 4 | 2.33 (1) | 0.004 | 2.4 (1.1) | 1.2 (1) | 995 | 0.047 | [3] |
| | | | Y-O$_2$ | 4 | 2.40 (1) | 0.004 | | | | | |
| | | 2nd | Y-P | 2 | 3.04 (1) | 0.007 | | | | | |
| | | | Y-Y | 4 | 3.79 (1) | 0.007 | | | | | |
| | | | Y-O$_1$-O$_2$ | 16 | 3.87 (1) | 0.009 | | | | | |
| | | | Y-O$_3$ | 4 | 4.19 (2) | 0.007 | | | | | |
| | | | Y-O$_4$ | 8 | 4.26 (2) | 0.007 | | | | | |
| Eudialyte | Kringlerne, Ilímaussaq, Greenland | 1st | Y-O$_{1a}$ | 2 | 2.24 (1) | 0.004 (1) | 1.2 | 1 | 33.4 | 0.017 | [4,5] |
| | | | Y-O$_{1b}$ | 4 | 2.30 (1) | 0.004 (1) | | | | | |
| | | 2nd | Y-Fe | 1 | 3.29 (6) | 0.007 (6) | | | | | |
| | | | Y-O$_{2a}$ | 2 | 3.35 (1) | 0.004 (1) | | | | | |
| | | | Y-Na | 2 | 3.43 (3) | 0.006 (4) | | | | | |
| | | | Y-Si | 6 | 3.50 (3) | 0.006 (4) | | | | | |
| | | | Y-O$_{2b}$ | 2 | 3.53 (1) | 0.004 (1) | | | | | |
| | | | Y-Ca | 2 | 3.80 (5) | 0.008 (6) | | | | | |

EXAFS fits based on crystal structures from (1) Ni et al.[71], (2) Santos et al.[72], (3) Ni et al.[73], (4) Johnsen[74], (5) Borst et al.[40]. CN is coordination number. R is radial distance in ångström. Definitions of other fit parameters listed in this table are provided in the 'Data Processing' section of the methods. Errors shown within parentheses are on last digit of the fitted variable. Values without errors were fixed in the final fit.

**Table 2 EXAFS fitting results for the coordination state of clay-hosted Y in the regolith samples.**

| Sample | Depth | Shell | Path | CN | $R$ (Å) | $\sigma^2$ (Å$^{-2}$) | $\Delta E_0$ (eV) | $S_0^2$ | $\chi^2$ | R-factor | n |
|---|---|---|---|---|---|---|---|---|---|---|---|
| Zhaibei, China | | | | | | | | | | | |
| LJ316 | 2 m | 1st | Y-O | 8.3 (9) | 2.38 (1) | 0.007 (2) | −1.5 | 1 | 317 | 0.017 | 16 |
| Ambohimirahavavy, Madagascar | | | | | | | | | | | |
| PIT2 | 4.8 m | 1st | Y-O | 7.9 (8) | 2.37 (1) | 0.007 (2) | −1.5 | 1 | 120 | 0.013 | 14 |
| MAD238-217 | 9 m, 4 m | 1st | Y-O | 8.1 (8) | 2.35 (1) | 0.008 (2) | 0 | 1 | 11 | 0.018 | 5 |

CN is coordination number. R is radial distance in ångström. n is number of merged spectra. Definitions of other fit parameters listed in this table are provided in the 'Data Processing' section of the methods. Errors shown within parentheses are on last digit of the fitted variable. Values without errors were fixed in the final fit.

siloxane complexes [≡Al-O-Ln]$^+$ and [≡Al-O-Ln(OH)$_2$]) would start to form at pH 6 (and less for HREEs) and possibly as low as pH 5. However, Malagasy pore waters measured during fieldwork range from pH 6.98 to 8, and hence, at close to neutral pH and low ionic strength, REEs are likely to be present as eight- and nine-coordinated hydrated outer-sphere basal surface complexes (Fig. 8). At higher pH, and at higher ionic strength, desolvation of the Ln$^{3+}$ ions may lead to the formation of inner-sphere complexes, either on the aluminol or siloxane surfaces or terminal OH groups. This finding is significant as the lack of hydrolysis and the presence of hydrated Ln$^{3+}$ ions adsorbed to the clay surface is critical in the ease of leaching and hence the economics of these deposits. Notably, the EXAFS data show that the coordination state of Y$^{3+}$ is identical within the Chinese and Malagasy laterite samples that are rich in exchangeable HREEs, and hence we infer a common REE adsorption mechanism across the deposits.

Our data do not clearly differentiate between kaolinite and halloysite, but previous studies suggest that halloysite has an enhanced REE adsorption capacity within weathering profiles[30,31,36,55,56]. Given that kaolinite is dominant over halloysite in the Malagasy samples, and halloysite (10 Å) is not distinctly identified in the Chinese sample studied here, we suggest that the majority of REE$^{3+}$ is adsorbed as 8–9-fold hydrated outer-sphere complexes to kaolinite. While halloysite typically has a higher porosity and specific surface area than kaolinite[65], recent experiments by Yang et al.[56] have shown that halloysite has a lower specific surface area normalised adsorption capacity for the REEs than kaolinite under the same conditions of pH and ionic strength. This may be because the variable charge aluminol surfaces on which REE$^{3+}$ adsorption is inferred to take place is inside the halloysite tubes rather than on the outer surface[65]. The respective roles of halloysite and kaolinite within regolith-hosted REE deposits and their progressive transformation during weathering are still not fully understood and remain an important subject of further study.

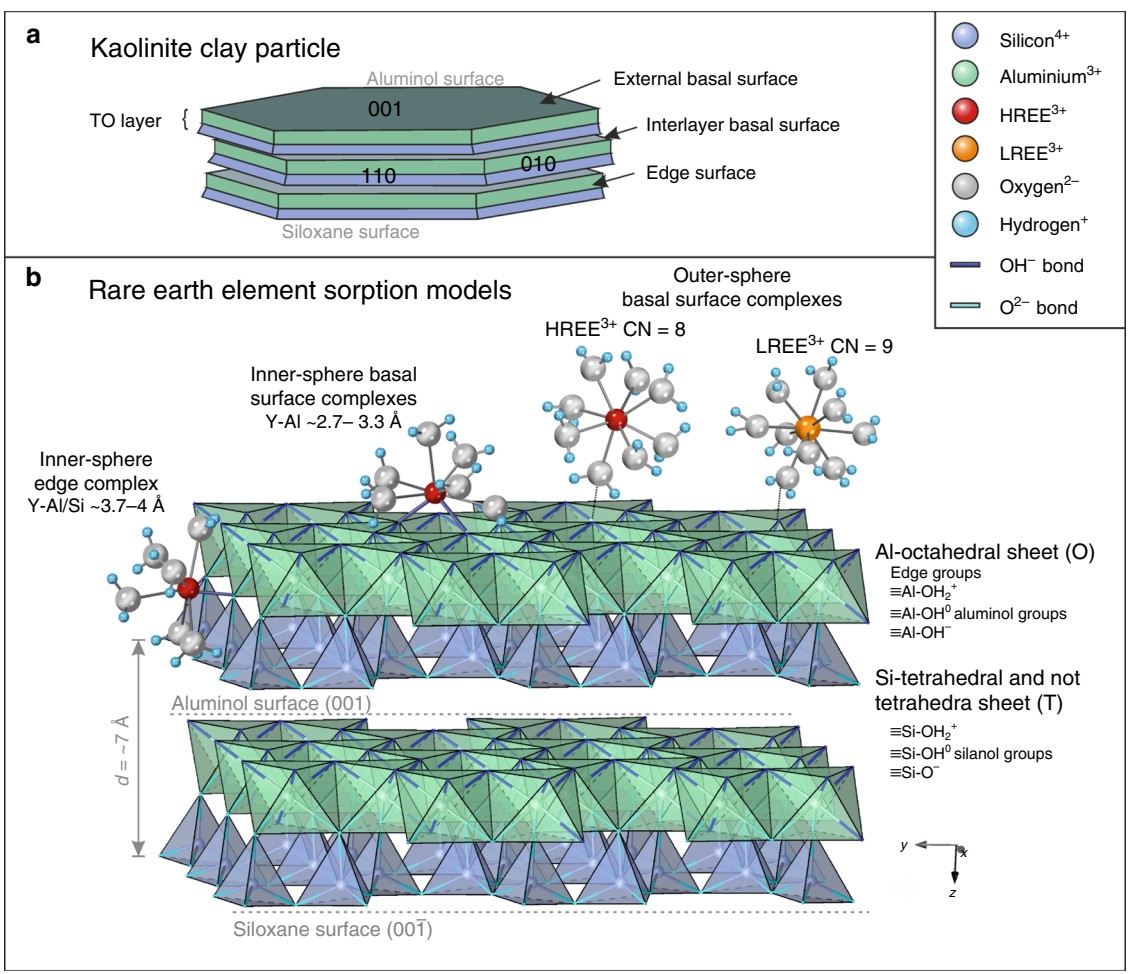

**Fig. 8 Schematic adsorption model of rare earth elements onto kaolinite. a** Schematic kaolinite structure showing the 1:1 stacking of Al-octahedral (O) and Si-tetrahedral (T) sheets, forming TO layers with aluminol and siloxane external basal surfaces, edge and interlayer surfaces. **b** Rare earth sorption model associated with Al-octahedral (O) sheets of kaolinite. Light and heavy rare earth element (LREE$^{3+}$ and HREE$^{3+}$) complexes may occur as eight- to nine-coordinated inner-sphere (basal surface or edge) complexes via mono- or bidentate bridging oxygens, or loosely adsorbed as 8- or 9-fold hydrated outer-sphere complexes. Our data demonstrate that Y in the studied ion-adsorption ores are dominantly present as 8-fold hydrated outer-sphere basal surface complexes. The data show no evidence for Al or Si scattering within a distance of 4 Å, thereby excluding the inner-sphere adsorption model. A similar sorption model can be made for halloysite (7 and 10 Å), which typically displays a tubular morphology with the aluminol basal surface on the inside of the tubes[65]. CN coordination number.

Of additional importance for understanding both deposit genesis and development of REE-leaching strategies for laterite deposits is that our XANES and EXAFS studies capture the easily leachable fraction of the REEs[63]. To ensure that we have measured the easily leachable REE fraction associated with clay minerals in our samples, we also measured Y K-edge XANES of bulk powders prepared from the laterite samples, pre- and post leaching with $(NH_4)_2SO_4$ (Supplementary Data 2), and compared these to leachates of an IMERYS kaolinite standard doped in synthetic REE solutions (Supplementary Figs. 5–7, Supplementary Table 2). This also allowed us to test for changes in the structural state of REEs during sample processing to produce polished thin sections. The XANES spectra for the bulk powders are shown unnormalised in Supplementary Fig. 6 to illustrate the reduction in Y K-edge absorption intensity before and after leaching. The markedly lower absorption of the post-leach material demonstrates that the XANES acquired for the pre-leach material are characteristic of the REEs in leachable form. We are therefore confident that we are directly characterising the structural state of the leachable; hence, economic fraction of the REEs in this deposit type. Moldoveanu and Papangelakis[63,64]

modelled the desorption behaviour of REEs during leaching as substitution of dehydrated monovalent cations for the hydrated Ln$^{3+}$ at the clay surface. Because monovalent cations can dehydrate completely at ambient temperature, they form a strong ionic bond with the clay surface, compared to the weak adsorption of the hydrated lanthanides. Our data are fully consistent with this model and support further development of simple, environmentally friendly, leach solutions based on this mechanism.

Supplementary Figure 7 also shows the normalised XANES of the prepared clay samples pre- and post leaching. The post-leach XANES demonstrate a minor peak at 17,065 eV (Feature B), which resembles spectral features of the high point-symmetry standards (e.g. zircon, Fig. 5). These features, typical of high symmetry site occupations, are consistent with the non-leachable REEs being structurally bound to kaolinite, in interlayer positions or as partially desolvated inner-sphere complexes. Inner-sphere complexation may also explain the anomalous XANES for kaolinite with low or background Y concentrations in the Malagasy PIT2 pedolith sample (Fig. 5b), which shows similar spectral features as the post-leach XANES spectrum (Supplementary Fig. 7).

Analysis of saprolite and saprock samples in the Madagascar laterite profiles highlights the importance of the primary (magmatic and hydrothermal) REE-hosting phases in the genesis of IADs. The complex controls of bedrock mineralogy in the Ambohimirahavavy complex on the release of REEs into the regolith profile were explored in more detail by Estrade et al.[28]. They showed that easily weatherable zirconosilicates, such as EGMs, are dominant primary REE hosts in the bedrock, and that REE enrichment within the pedolith and saprock is correlated to the local occurrence of REE-rich pegmatite veins containing EGMs. In places, late-magmatic hydrothermal breakdown of EGMs resulted in the formation of secondary zircon and zirconolite (Fig. 3d), which are more resistant to weathering and subsequently inhibit the mobility and leaching of HREEs into the laterite profile. The latter phases are clearly identifiable by high symmetry Y XANES of microscale Y-rich reaction products. This is consistent with previous studies demonstrating that zircon is resistant to weathering[12,16,28], but contradicts the conclusion of Ram et al.[31] that zircon breaks down completely in the weathering profile and forms an important source of REEs (and Zr) that are subsequently adsorbed to clays.

Differences in bedrock mineralogy provide the main distinction between the Malagasy and Chinese samples, and this is reflected in the complexity of relict mineral phases that accommodate the REEs in the Malagasy saprolite samples. In the Zhaibei granite, primary Zr phases are less prevalent than in the Madagascar samples, and a higher proportion of the REEs, particularly HREEs, are released from more easily weatherable REE phases (e.g. biotite, monazite, Ca-REE fluorcarbonates, fergusonite-(Y), aeschynite-(Y)) and subsequently adsorbed to clay minerals in the weathering profile. The observation of the partial breakdown of pyrochlore-group minerals to form REE fluorcarbonates in the Madagascar samples (Fig. 3e, f) indicates that the more reactive niobates may also be an important source of the REEs in weathering solutions.

In summary, economically exploited IADs on the Zhaibei granite in China were compared to prospective regolith profiles on peralkaline granites and syenites of the Ambohimirahavavy complex in Madagascar. Ammonium sulfate leaching studies demonstrate high levels of exchangeable REEs associated with clay minerals, dominantly kaolinite, in both Chinese (<1000 p.p.m. TREEs, incl. 29–44% HREEs) and Malagasy samples (up to 1963 p.p.m. TREE, incl. 6–29% HREEs). Petrographic characterisation combined with μSXRF mapping show that REEs are released from primary REE-hosting phases in the bedrock, such as pyrochlore, eudialyte and zircon, during late-magmatic alteration and subsequent supergene weathering. Y and Nd XANES for the Chinese and Malagasy clay-hosted REEs are comparable to XANES of Y in aqueous solution and Nd in REE fluorcarbonates, in which Y and Nd are eight and nine coordinated, respectively. The structural state of Y associated with kaolinite was further constrained using EXAFS, yielding coordination numbers (CNs) of 7.9–8.3 ± 0.9 with average Y-O bond distances of 2.35–2.38 ± 0.01 Å. The X-ray absorption data, combined with leaching experiments, confirm that a significant proportion of the light HREE and HREE are genuinely adsorbed to clay mineral surfaces. We show that REE-adsorbed clays from Madagascar and China are direct structural analogues in which the REEs occur dominantly as easily exchangeable eight to nine coordinated, hydrated, outer-sphere, basal surface complexes adsorbed to kaolinite (and in the case of Madagascar, minor halloysite), rather than as inner-sphere or interlayer complexes. Our data thus explain the easily leachable nature of these economically important ore types and suggests a common adsorption mechanism across both deposit sites. The identification of a common mechanism at two geographically separated sites

confirms that this process operates globally in lateritic weathering profiles and further supports the search for critical REE deposits in the supergene environment.

## Methods

**Sample and standard preparation.** Samples were prepared as polished thin sections (Madagascar samples) or as polished resinated regolith mounts (LJ316, China). Thin sections were produced at the British Geological Survey (Keyworth, UK), from either intact blocks of saprock and saprolite or from surface pedolith samples collected using soil boxes. Samples were first impregnated with resin and then mounted on spectroscopically pure glass slides prior to polishing with diamond paste, in order to prevent interference on the XANES from trace elements incorporated in the glass slide. Disaggregated materials from Chinese samples were mounted in resin and then also polished. Cerium oxide polishing media were avoided. A powdered fraction of all samples prepared as resin-impregnated polished blocks or sections was also prepared as a pressed powder pellet with no further treatment in order to ensure that the process of resin mounting did not modify the clay mineralogy or REE coordination. Details of the sample materials used are provided in Supplementary Data 1.

Analytical standards and blanks measured during the XAS studies were prepared as powders from natural and synthetic REE minerals in which the REEs occupy different coordination states (Supplementary Data 1). Microcrystalline Y-doped $NdPO_4$ was synthesised at the University of St. Andrews following procedures described in Borst et al.[40]. Synthetic crystals of Nd-doped $YPO_4$ were provided by Lynn Boatner, ORNL[66]. All analytical standards were ground under ethanol in an agate mortar and checked for phase purity by powder XRD (Philips 1050 XRD instrument with monochromated Co-Kα-radiation, University of St. Andrews) prior to X-ray absorption measurements.

Single cation Y and Nd solutions were prepared from the oxide dissolved in 5% nitric acid ($HNO_3$) (Inorganic Ventures CGY1–1000 p.p.m. Y in nitric acid, and CGNd1—1000 p.p.m. Nd in nitric acid), alongside a synthesised REE-bearing kaolinite standard at the University of Brighton (UK). The kaolinite was prepared using IMERYS reagent grade "light kaolinite", which was analysed by inductively coupled plasma mass spectrometry (ICP-MS) prior to experiments to identify any initial REE content. The exchangeable REE fraction was then leached with ammonium sulfate to remove any surface adsorbed cations, and then saturated for 16 h with 30 p.p.m. REE solution with 0.1 M $NaNO_3$ (prepared from Agilent 8500-6944 100 p.p.m. REE oxides in 5% $HNO_3$) and dried. A split from the re-adsorbed kaolinite sample was leached with ammonium sulfate, and the bulk material was fused with lithium borate and digested in nitric acid to confirm the mineral framework REE content, and ion-exchangeable REEs adsorbed to the mineral surfaces. The leach and experimental solutions were analysed for REE contents by ICP-MS (Agilent 7900) at all intermediate stages. The octopole reaction system was operated in He mode to reduce polyatomic interferences. The samples and a matrix-match blank solution were diluted with 2% $HNO_3$, and the blank was used to calibrate the REEs from 1 to 500 μg L$^{-1}$ using the multielement calibration standard (8500-6944 Agilent). Internal standard (Rh) was added automatically to all standards, samples and blanks via the online internal standard solution and measured continuously alongside sample analysis. REE contents for each step in the IMERYS kaolinite leaching and adsorption experiments are shown in Supplementary Fig. 4 and the data provided in Supplementary Table 2.

Following X-ray absorption measurements, corresponding areas on the thin sections (Madagascar) and resin mount (China) were characterised and imaged by SEM at the University of Brighton using a Zeiss EVO LS 15 SEM equipped with an Oxford Instruments XMax 80 EDX spectrometer, at 20 kV accelerating voltage and 1.2 nA beam current. Samples were carbon coated to avoid charging, so all SEM analyses were carried out to confirm mineralogy of the studied areas after XAS in order to avoid sample contamination issues. Blank analyses were also acquired on areas of just glass and slide preparation resin in order to demonstrate no interference from the sample preparation media.

**Clay characterisation.** The clay minerals were characterised using powder XRD at the University of Brighton (UK) using a Panalytical MRD X'pert Pro High Resolution powder diffractometer in Bragg-Brentano geometry using Ni-filtered Cu-Kα radiation. Tube conditions were 40 kV and 40 mA. Randomly oriented powders were prepared using ~10 g of dried bulk sample which was manually ground in an agate mortar to produce a homogeneous fine powder. The fine powder was then pressed into a steel ring to get a mechanically stable sample and a flat surface. The mounts were typically X-rayed from 5° to 70° 2θ at a scanning rate of 0.01° min$^{-1}$. Detection limit is ~2% of the sample. The clay fraction of selected samples was separated and further characterised. The clay fraction was isolated by repeatedly mixing a few grams of sample with a solution of deionised water saturated with a dispersive agent (Na-hexametaphosphate). The suspension was treated for 120 s with an ultrasonic bath to increase clay dispersion and centrifuged at 750 r.p.m. for 260 s. The supernatant containing the clay fraction was collected and centrifuged at 3500 r.p.m. for 30 min. The supernatant was then discarded and a few drops of deionised water were added to the clay concentrate. Slides of well-oriented clay were made by putting a few drops of the clay suspension on glass slides, which were

then air-dried. Results for LJ316 are shown in Supplementary Fig. 2 and SEM images shown in Supplementary Fig. 3.

**Leaching experiments.** Leaching studies in support of the X-ray absorption study were carried out on bulk powders prepared from individual samples (LJ316, PIT2, MAD2341, MAD219, Supplementary Table 1). One gram of each sample was dried in an oven (105 ± 2 °C) until constant weight and mixed with 40 mL of 0.5 M $(NH_4)_2SO_4$ solution[64], adjusted to pH 4 with $H_2SO_4$, in a 50 mL centrifuge tube. The mixture was immediately shaken at 30 ± 10 r.p.m. for 1 h using a mechanical end-over-end shaker at room temperature. The extract was separated from the solid residue by centrifuging at $3000 \times g$ for 20 min. The solution obtained was filtered through 0.22 μm syringe filters and the filter was washed with 50 mL deionised water. Samples were stored in a fridge at 4 °C prior to analysis. For each batch of 14 samples, a blank was prepared following the same procedure. No certified reference material exists for REE-leaching experiments, but repeat analyses of BCR701[67] showed that leaches reproduced values within 5% precision for transition metals.

**X-ray absorption spectroscopy.** XAS data were collected at the I18 microfocus beamline at Diamond Light Source Ltd (Didcot, UK), a 3 GeV third-generation synchrotron facility typically operating at a current of 300 mA. I18 is designed for high spatial resolution analyses of heterogeneous samples within the 2–20.7 keV energy range, and is set up for μSXRF mapping, μXRD and μXAS[68]. All measurements were run at room temperature conditions (~295 °K), using a $3 \times 3$ μm focussed X-ray beam. Standards were measured as powders or crystals (gem-quality diamond and zircon) mounted on KAPTON© tape, and Y and Nd aqueous solutions were held in a metallic liquid cell sealed between KAPTON© tape. Absorption spectra for a subset of the REE-bearing mineral standards are reported in Borst et al.[40]. Samples and standards with medium to low Y and Nd contents were measured in fluorescence mode, and REE-rich mineral standards were measured in transmission mode. The energy range was set to include both the XANES and EXAFS regions for the Y K-edge (17038 eV) and the Nd $L_3$-edge (6208 eV). Yttrium was selected as a proxy for HREE as it can be analysed on the K-edge, avoiding significant interference from absorption edges of other elements over the measured energy range. Nd was chosen because it is of key economic interest and because natural materials contain no Pm, which allows for a longer interference-free EXAFS energy range than other lanthanides. The Y XANES, Nd XANES and Y EXAFS data are provided in Supplementary Data 2, Supplementary Data 3 and Supplementary Data 4. The raw synchrotron data and supporting metadata can be accessed from[69] respectively.

Blank XAS analyses of glass and resin showed no REE adsorption edges, and hence no interference on the measurements of laterite material. A diamond procedural blank gave no edge demonstrating no contribution to the spectra from scattered X-rays encountering the specialist metals of the stages and beamlines. In order to check for any changes in the structural state of kaolinite or the coordination of the REEs during sample preparation, spectra were also collected from bulk powdered samples of the studied laterites, which were dried and pressed into pellets, but with no other sample preparation. Yttrium K-edge and Nd $L_3$-edge XANES from these samples were indistinguishable from the XANES collected from polished thin sections and blocks indicating no influence on the measurements from the sample preparation (Supplementary Data 2, Supplementary Data 5).

Energy calibration of the Nd $L_3$-edge spectra (6208 eV) and Y K-edge spectra (17,038 eV) was done by measuring the Mn K-edge (at 6539 eV) and Zr K-edge (at 17,998 eV) from Mn and Zr metal foils, respectively. Detector count rates were checked to ensure measurement within the linear range. The width of the energy interval defining elemental peak areas was optimised to exclude influence from fluorescence signals of neighbouring elements. The Nd $L_3$-edge XAS region also contains the Ce $L_2$-edge at 6164 eV (in the pre-edge region, Fig. 6) and the Pr $L_2$-edge at 6440 eV in the EXAFS region. Tight windowing of the secondary Nd Lα X-rays (minimising the response from Ce $L_2$ and Pr $L_2$) and insertion of Al foils in the path of the X-ray beam to filter out lower energy X-rays were used to minimise $L_2$-edge absorption peaks for Ce and Pr. However, interference of both Ce $L_2$ and Pr $L_2$ reduces the useable Nd spectrum to a k-range of 2–9 Å$^{-1}$ and this subsequently restricts the number of independent variables in the EXAFS fitting procedure. Hence, we only present the Nd $L_3$-edge XANES. The μSXRF element maps were collected at an incident beam energy of 18.2 keV using a focussed beam of c. $3 \times 3$ μm$^2$, a 5 μm step size and 0.5 s dwell time per pixel. False colour element maps were produced by fitting the fluorescence peaks for Y, Fe and Mn from the full fluorescence spectra using The Data Analyses Workbench (DAWN) and PyMCa software.

**Data processing.** Processing of the Y K-edge and Nd $L_3$-edge XAS data was performed using the Athena and Artemis Demeter Perl software packages (version 0.9.25[45]). The XAS spectra were normalised to the X-ray source intensity ($I_0$) and the absorption edge step height to yield $\mu(E)$. The absorption edge was set to the maximum of the first derivative. Backgrounds were removed using the AUTOBK background subtraction algorithm[70], using spline k-ranges of 0–11.8, an $R_{bkg}$ value of 1.0, and avoiding the Ce $L_2$-edge absorption peak in the pre-edge of the Nd $L_3$-edge spectra. Multiple runs were merged to improve signal-to-noise ratios. The Y

K-edge EXAFS curves were analysed and fitted to theoretical scattering paths calculated from crystallographic models using the FEFF6 program built into Artemis[45]. For standards with multiple coordination spheres or scattering paths, the spectra were fitted in a shell-by-shell fashion (i.e. shells in the Fourier transform radial distribution functions) to construct crystallographically realistic fitting models. For the samples, the fitted k-range and R-range was 2.5–10 and 1.2–3 Å, respectively, to yield compatible fitting parameters (P) and a consistent number of independent parameters ($2\Delta k\Delta R/\pi + 1$) supported by the data. The quality of the fit is assessed by the R-factor and the reduced $\chi^2$ function calculated in Artemis. Fits of acceptable quality yield R-factors of 0.02 or below[45]. Refined parameters include the energy offset ($\Delta E_0$ in eV), CNs, the mean interatomic distance between the absorber and the scattering atoms ($R_{Y-O}$, $R_{Y-F}$, $R_{Y-P}$ in Å) and mean-square relative displacement Debye–Waller factors per scatterer ($\sigma^2$ in Å$^{-2}$). The amplitude reduction factor ($S_0^2$) was fixed to 1, which was determined from a three-shell fit of the $Y_2O_3$ reference powder (see ref. [40]). Initial first-shell refinements with four variables were evaluated and improved iteratively and the robustness and codependence of the fitted parameters checked by systematically fixing and changing one of the variables. The final refinements of the kaolinite and Y solution spectra are fitted for two variables (CN and $\Delta R$), where $\Delta E_0$ and $\sigma^2$ values are fixed to best-fit values obtained in the previous fits (Tables 1 and 2). The $k^2$-weighted Y EXAFS spectra and fitted models are provided in Supplementary Data 4. Underlying Artemis files for the final fits are provided in Supplementary Data 6.

## Data availability
The authors declare that all data supporting the findings of this study are available within the paper and its supplementary files. Unprocessed data can be accessed from the University of St Andrews Research Portal, https://doi.org/10.17630/73e556f7-0fc3-4814-b25c-fa747ceb72a2[69].

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

## Acknowledgements

This work was supported by the NERC SOSRARE consortium, grant numbers NE/M010856/1 (to A.M.B., A.A.F. and N.J.H.), NE/M011267/1 (to M.P.S., G.E., P.N., C.V.-d.-B. and E.M.) and NE/M01116X/1 (to K.M.G.). J.K. was supported by The Czech Science Foundation GACR EXPRO (grant number 19-29124X). We thank Diamond Light Source for beam time at the I18 beamline (grants SP14793 and SP15903). Rocky Lowell Rakotoson and Fetra Rasolonirina are thanked for their invaluable assistance in organising fieldwork in Madagascar. Tantalus Rare Earths AG provided assistance with field logistics, and access to drill core. We thank Lynn Boatner (ORNL, USA), Henrik Friis (NHM Oslo), National Museum Scotland, the Hunterian Museum and SOSRARE partners for providing mineral standards.

## Author contributions

A.M.B., M.P.S., A.A.F. and P.N. planned the project and performed synchrotron analyses at Diamond Light Source with the assistance of beamline scientist K.G. Fieldwork and sampling in Madagascar was undertaken by M.P.S., G.E., K.M.G. and E.M. Chinese samples were collected and prepared by M.P.S., C.X. and J.K. Mineral standards were prepared by A.M.B. and N.J.H. at St. Andrews. Clay samples and standards prepared by M.P.S. and P.N. at Brighton. M.P.S., G.E., E.M. and C.V.-d.-B. performed SEM, XRD and ICP-MS analyses. A.M.B. processed the X-ray absorption data and produced figures. A.M.B. and M.P.S. wrote the manuscript with input from all others.

## Competing interests

The authors declare no competing interests.
