## [Peer Review File · Nature Communications]

Reviewers' comments:

Reviewer #1 (Remarks to the Author):

This is an excellent contribution to advancing our understanding of the complexities in micro to nanoscale processes that lead to formation of REE ion adsorption clay resources, in particular the heavy REE types. The Authors have included significant data and thoughtful interpretations.

1. Consider adding this reference to Southeastern US regolith-related IAC REE deposits: Foley, N.K., and Ayuso, R., 2015. REE enrichment in granite-derived regolith deposits of the southeastern United States: Prospective source rocks and accumulation processes. In: Simandl, G.J and Neetz, M., (Eds.), Symposium on Strategic and Critical Materials Proceedings, British Columbia Geological Survey Paper 2015-3, pp. 131-138.

2. Carefully review the text for minor typos, a few repeated words (e.g., line346), etc.

Reviewer #2 (Remarks to the Author):

Overall, I commend the authors of this paper on a thorough analysis. I question the novelty of the research and have suggested potential avenues in which to further enhance the readability of the manuscript. My detailed comments are provided as an attached document.

DECISION: ACCEPT WITH MAJOR REVISIONS

Structural State of Rare Earth Elements in Regolith-Hosted Clay Deposits from China and Madagascar

The present paper compares the structural state of REE in Chinese clays to REE in prospective lateritic clays from Madagascar. We use synchrotron X-ray Absorption Spectroscopy and X-Ray Fluorescence to map REE distributions and determine the bonding environment of Y, as a proxy for HREE, in the ore. Additionally, they use Nd as a proxy for LREE to determine the association/dissociation thereof of LREE vs. HREE in these ion-adsorption clays (IACs).

Their primary result indicates that HREE are enriched within clay minerals and absorbed as easily leachable 8-coordinated outer-sphere hydrated complexes. Additionally, they highlight that, at the atomic level, HREE are truly adsorbed and the Malagasy clays are genuine mineralogical analogues to those currently mined in China.

Overall, while I commend the efforts of the authors in comparing the primary regolith-hosted clay deposits of China (Yiangxi Province) with those of the Madagascar province (Ambohimirahavy complex); the results do not provide significant new insight into these IAC deposits. Therefore, I do not recommend its publication in its present form and I look forward to a revised manuscript for further perusal. A more detailed list of my comments is provided below:

Line 31-33: “Similar deposits elsewhere might provide alternative supply for these high-tech metals, but the nature of the clays is unclear and the adsorbed state of REE within them never demonstrated.”

This is untrue; the nature of REE is well understood to be absorbed as primarily +3 oxidation state onto these aluminosilicate minerals. Further, the oxidation of Ce^{3+} to Ce^{4+} in the presence of Mg/Fe bearing minerals within the deposit and subsequent precipitation as cerianite is also well established. The seminal works of Kynicky et al., 2012, Xu et al. 2017 and Moldoveanu and Papangelakis, highlights this effectively.

Perhaps better context and clarity is required to properly frame this work to encapsulate previous work in this area.

Kynicky, J., Smith, M.P. and Xu, C., 2012. 'Diversity of Rare Earth Deposits: The Key Example of China'. Elements, 8: 361-67.

Xu, C., Kynicky, J., Smith, M.P., Kopriva, A., Brtnický, M., Urubek, T., Yang, Y., Zhao, Z., He, C. and Song, W., 2017. 'Origin of heavy rare earth mineralization in South China'. Nature communications, 8: 14598.

Moldoveanu, G.A. and Papangelakis, V.G., 2012. 'Recovery of rare earth elements adsorbed on clay minerals: I. Desorption mechanism'. Hydrometallurgy, 117: 71-78.

Moldoveanu, G.A. and Papangelakis, V.G., 2013. 'Recovery of rare earth elements adsorbed on clay minerals: II. Leaching with ammonium sulfate'. Hydrometallurgy, 131: 158- 166.

Line 46: Mild acidification in this case refers to Ammonium sulfate leaching of these IAC's which is currently practised in China. The environmental implications of this (stripping large areas, extensive groundwater contamination) should be noted as an added point.

Line 46-47: “This property led to the assumption that the REE within them are weakly adsorbed onto clay mineral surfaces, even though such behaviour is also consistent with e.g. the dissolution of nanoparticulate secondary minerals.”

This statement requires further clarification. The current assumption is that the three main types of REE occurrence on the hydrated aluminosilicate are (i) ion exchange phases, (ii) colloid phases and (iii) mineral phases. Therefore, the distribution and chemical state of REEs in IAC deposits can be complex. A significant fraction of the REE budget is available via ion-exchange: these REEs are adsorbed on a number of active sites, including: “Broken edge” sites and exposed surface aluminol and silanol groups; hydrophilic silanol surfaces; hydration shells of exchangeable cations; hydrophobic sites on adsorbed organic molecules; isomorphic substitutions (e.g. Al³⁺ for Si⁴⁺); and substitution of exchangeable cations (K⁺, Ca²⁺) (Schoonheydt and Johnston, 2011; Zhou and Keeling 2013). Non-ion-exchange REE forms include colloids deposited as insoluble oxides, hydroxides or polymeric organometallic compounds; mm-to µm-size REE- rich secondary mineral phases such as cerianite-(Ce); and residual refractory magmatic REE- rich minerals (Joussein et al. 2005)

Schoonheydt, R.A. and Johnston, C.T., 2011. 'The surface properties of clay minerals'. In Layered Structures and their Application in Advanced Technologies, EMU Notes in Mineralogy, 11: 337-373.

Zhou, C.H. and Keeling, J., 2013. 'Fundamental and applied research on clay minerals: from climate and environment to nanotechnology'. Applied Clay Science, 74: 3-9.

Joussein, E., Petit, S., Churchman, J., Theng, B., Righi, D. and Delvaux, B., 2005. 'Halloysite clay minerals- a review'. Clay Minerals, 40: 383-426.

Line 77: “Such studies therefore do not constrain the speciation of the HREE or allow for determination of the behaviour of the economically most significant elements.”

The authors are correct in this assessment. However, this is partly due to the key focus on Ce partitioning within the IACs. This forms the novel research provided in the present study. The focus on Y speciation compared to Nd offers the first look at HREE absorption/complexation within IACs. This point can be better emphasised.

Figure 1: The authors need to add the IAC deposits of Philippines. The prospects of N.Vietnam are also not properly shown.

Sanematsu, K. and Watanabe, Y., 2016. Characteristics and genesis of ion adsorption-type rare earth element deposits. Rev. Econ. Geol, 18, pp.55-79.

Padrones, J.T., Imai, A. and Takahashi, R., 2017. Geochemical behavior of rare earth elements in weathered granitic rocks in northern Palawan, Philippines. Resource Geology, 67(3), pp.231-253.

Line 131: There is a sentence missing. I am assuming it is minor 7 Å halloysite (through dehydration of 10 Å halloysite)?

Line 138: The authors provide no mention of the XRD data for the Ambohimirahavavy complex, but in S2 make a note that the samples are distinct from the Malagasy samples in that 10 Å halloysite is largely absent. They refer to Estrade et al. 2019, but should make a mention of halloysite and other major/minor mineralogy since they report it for Zhiabei (given its importance that is discussed further below).

Figure 3 can be modified to differentiate between kaolinite and halloysite. Changing contrasts in the SEM shows brighter/darker zones easily observable. The zonation is somewhat observed in Fig 3f. Optical microscopy can also show this.

Figure 4: “The ‘hotspots’ of Y associated with clay minerals occur where no distinct bright REE-rich phases are identified in backscatter and EDS analyses. Element maps for the Malagasy samples with the highest leachable HREE fractions show similar elemental distributions, with Y visibly enriched around the edges of clay minerals.”

The authors need to provide more information here. They only show Y maps with Fe and Mn. The SXRF should also have Ce and La (maybe Nd?) maps which also need to be included. They will show ubiquitous distribution of the LREE which follow a similar trend to the Y. However, they also note higher Y visibly enriched around the edges of clay minerals. Ram et al. 2019 reports that “while the majority of the Y is homogeneously distributed throughout the clay, i.e. likely sorbed as trivalent Y^{3+} ; there is a preferential enrichment in the white phase (consisting mainly of halloysite) relative to the tawny phase (kaolinite).” This is an important distinction with the Malagasy samples, and is worthy of further characterisation to resolve the current data to their association within individual clay fractions. In the case of halloysite, the tubes are highly disordered, ranging in length from 1 μm to 15 μm . Joussein et al. (2005) reported evidence of these tubular morphologies/fibrous structures that are commonly bent and irregular, and could likely be a source of higher adsorption of Y.

Ram, R., Becker, M., Brugger, J., Etschmann, B., Burcher-Jones, C., Howard, D., Kooyman, P.J. and Petersen, J., 2019. Characterisation of a rare earth element-and zirconium-bearing ion-adsorption clay deposit in Madagascar. Chemical Geology, 522, pp.93-107.

Joussein, E., Petit, S., Churchman, J., Theng, B., Righi, D. and Delvaux, B., 2005. 'Halloysite clay minerals- a review'. Clay Minerals, 40: 383-426.

Y K-edge XANES

The authors report: “They compare most closely to XANES of parisite-(Ce) and Y in solution, in which Y occupies 9 and 8-fold coordinated sites, respectively, and existing as 3+ oxidation state. The XANES of the 8-coordinated high symmetry standards (zircon and xenotime) also show Y as 3+ oxidation state. The spectra show a systematic shift in the position and shape of the broad peak around 17104 eV (Feature C), shifting to higher energy with decreasing coordination numbers (from 17104 eV for Y in low-symmetry 11 to 8-fold sites, to 17110 eV for phases with Y in 6-coordinated sites.”

So, am I correct in the assessment that this proves that Y exists within the zircon lattice and hosted within the zircon rather than co-sorption of Zirconium alongside Yttrium? If so, this is an important distinction to be made, as previous SXRF has shown Zr to be ubiquitous with Y in distribution in the IAC's with localised hot spots associated with zircons. So clarity would be appropriate on whether all Y/Zr relationships exist primarily related to Zircons and not otherwise.

Nd L₃-edge XANES

The Nd XANES shows the Nd exists primarily in +3 oxidation state. Feature B varies significantly from Madagascar and China and is not mentioned properly here. This needs to be explored further. The variable Ce^{3+} to Ce^{4+} ratio is also dependent on Mn/Fe mineralogy which is not reported herein.

Coordination of Y (EXAFS)

Please include models of the localised bonding environment of Y within the spectra itself. This allows for easier readability. This can be done using easy to use software.

The primary result suggests that the 8-fold coordinated sites with respect to Y and O. When the authors discuss the aqueous phase of Y (and Nd) do they refer to a leachate solution of this ore, and synthetic solution interchangeably. They provide greater distinction of their conditions in S5 and S6 but will be worth some clarification in the main text.

Discussion

A significant portion of the discussion is well established knowledge; the nature of REE³⁺ adsorbed to IACs. See previous papers herein. However, a key distinction is between kaolinite and halloysite. Given the dewatering characteristics of kaolinite vs. halloysite, previous studies have shown greater sorption to halloysite. This needs to be properly reflected within the figure and discussion. The greater incidence of REE as adsorbed fraction is also well established and therefore its greater leachability. What this paper is missing significantly is providing context of the incidence of the REE (which is HREE dominated) between China and Madagascar and other such regolith-hosted clay deposits. For e.g. Ram et al. 2019 highlight the potential for Madagascar to be a viable non-chinese alternative to HREE through comparison of Ce anomaly and total REE+Y concentrations (See Figure below). While the current data show that REE-adsorbed clays from Madagascar and China are direct structural analogues in which the REE occur dominantly as easily exchangeable 8 to 9 coordinated, hydrated, outer sphere, basal surface complexes adsorbed to kaolinite (and possibly halloysite), rather than inner-sphere complexes, or interlayer complexes, the distinction between kaolinite and halloysite is certainly the novelty this paper can provide. Further comparison of these deposits to other deposits with data readily available can provide critical context in the search for critical REE deposits in the weathering environment.

[REDACTED]

Figure 10, Lam et al., Characterisation of a rare earth element- and zirconium-bearing ion-adsorption clay deposit in Madagascar, *Chemical Geology*, **Volume 522**, 20 September 2019, Pages 93-107

Response to Referees

27/04/2020

Reviewer 1

This is an excellent contribution to advancing our understanding of the complexities in micro to nanoscale processes that lead to formation of REE ion adsorption clay resources, in particular the heavy REE types. The Authors have included significant data and thoughtful interpretations.

We thank Reviewer #1 for these positive comments.

1. Consider adding this reference to Southeastern US regolith-related IAC REE deposits: Foley, N.K., and Ayuso, R., 2015. REE enrichment in granite-derived regolith deposits of the southeastern United States: Prospective source rocks and accumulation processes. In: Simandl, G.J and Neetz, M., (Eds.), Symposium on Strategic and Critical Materials Proceedings, British Columbia Geological Survey Paper 2015-3, pp. 131-138.

Reference included (Line 106)

2. Carefully review the text for minor typos, a few repeated words (e.g., line346), etc.

Line 346 corrected, and text checked throughout.

Reviewer 2

Overall, I commend the authors of this paper on a thorough analysis. I question the novelty of the research and have suggested potential avenues in which to further enhance the readability of the manuscript. My detailed comments are provided as an attached document.

We thank reviewer 2 for their helpful and constructive review. We now clarify the novelty of our work and take on board many of the reviewer's suggestions. A key point the reviewer had made relates to differentiating halloysite and kaolinite. We make clear in our reply that halloysite and kaolinite are in fact very difficult to distinguish using conventional techniques, it is certainly not possible to distinguish them in BSE images. The key point of our EXAFS work is that it demonstrates a similar adsorption mechanism for REE in both China and Madagascar materials, regardless of the fraction of halloysite present. We stress this in the manuscript and further below. Please find our detailed replies to the reviewer's comments below.

Attached comments

The present paper compares the structural state of REE in Chinese clays to REE in prospective lateritic clays from Madagascar. We use synchrotron X-ray Absorption Spectroscopy and X-Ray Fluorescence to map REE distributions and determine the bonding environment of Y, as a proxy for HREE, in the ore. Additionally, they use Nd as a proxy for LREE to determine the association/dissociation thereof of LREE vs. HREE in these ion adsorption clays (IACs).

Their primary result indicates that HREE are enriched within clay minerals and absorbed as

easily leachable 8-coordinated outer-sphere hydrated complexes. Additionally, they highlight that, at the atomic level, HREE are truly adsorbed and the Malagasy clays are genuine mineralogical analogues to those currently mined in China.

Overall, while I commend the efforts of the authors in comparing the primary regolith-hosted clay deposits of China (Yiangxi Province) with those of the Madagascar province (Ambohimirahavavy complex); the results do not provide significant new insight into these IAC deposits. Therefore, I do not recommend its publication in its present form and I look forward to a revised manuscript for further perusal. A more detailed list of my comments is provided below:

Line 31-33: “Similar deposits elsewhere might provide alternative supply for these high-tech metals, but the nature of the clays is unclear and the adsorbed state of REE within them never demonstrated.”

This is untrue; the nature of REE is well understood to be absorbed as primarily +3 oxidation state onto these aluminosilicate minerals. Further, the oxidation of Ce³⁺ to Ce⁴⁺ in the presence of Mg/Fe bearing minerals within the deposit and subsequent precipitation as cerianite is also well established. The seminal works of Kynicky et al., 2012, Xu et al. 2017 and Moldoveanu and Papangelakis, highlights this effectively.

Perhaps better context and clarity is required to properly frame this work to encapsulate previous work in this area.

Kynicky, J., Smith, M.P. and Xu, C., 2012. 'Diversity of Rare Earth Deposits: The Key Example of China'. *Elements*, 8: 361-67.

Xu, C., Kynicky, J., Smith, M.P., Kopriva, A., Brtnický, M., Urubek, T., Yang, Y., Zhao, Z., He, C. and Song, W., 2017. 'Origin of heavy rare earth mineralization in South China'. *Nature communications*, 8: 14598.

Moldoveanu, G.A. and Papangelakis, V.G., 2012. 'Recovery of rare earth elements adsorbed on clay minerals: I. Desorption mechanism'. *Hydrometallurgy*, 117: 71-78.

Moldoveanu, G.A. and Papangelakis, V.G., 2013. 'Recovery of rare earth elements adsorbed on clay minerals: II. Leaching with ammonium sulfate'. *Hydrometallurgy*, 131: 158- 166.

A number of studies of natural laterites and Ion Adsorption Deposits have concluded that 3⁺ oxidation state REE are adsorbed on to alumina-silicate minerals, most specifically on clays. This has been done from leaching data and comparison with experimental runs. ***No previous study has measured the structural state of the REE, except for Ce, in regoliths from sites of economic interest.***

Moldoveanu and Papangelakis (2012, 2013) published leaching studies to infer adsorption. We can comment particularly on the other two papers since they are from our group: Kynicky et al. (2012) was a review with no data to allow direct determination of the sites of the REE. Xu et al. (2017) was an isotope study, focussing on secondary REE minerals, with limited leaching data and no direct mineralogical information on the sites of REE within or on the surfaces of clay minerals. Studies of experimentally synthesised clays (extensively cited in the original manuscript) measured the co-ordination of 3⁺ cations on clay mineral surfaces by XAFS, but studies of natural systems have inferred the co-ordination from leaching data and not by direct measurement. We emphasise further this in the text.

We agree with the comment on Ce. Ce was not the focus of this study (a) because it has already been studied by XAFS in laterites (Janots et al., 2015; Ram et al., 2019), (b) it is not of economic interest in these deposits, and (c) its behaviour is decoupled from critical REE, as it is less abundant (as a result of the soil profile leaching processes) due to being retained as immobile Ce⁴⁺ in oxides (Cerianite – CeO₂). We clarify this point and further highlight

previous XAFS on Ce in the introduction. We also expand the discussion of the Ce absorption features in the Nd XANES section.

Line 46: Mild acidification in this case refers to Ammonium sulfate leaching of these IAC's which is currently practised in China. The environmental implications of this (stripping large areas, extensive groundwater contamination) should be noted as an added point.

We add a line and reference to note this issue, Line 67.

Line 46-47. “This property led to the assumption that the REE within them are weakly adsorbed onto clay mineral surfaces, even though such behaviour is also consistent with e.g. the dissolution of nanoparticulate secondary minerals.”

This statement requires further clarification. The current assumption is that the three main types of REE occurrence on the hydrated aluminosilicate are (i) ion exchange phases, (ii) colloid phases and (iii) mineral phases. Therefore, the distribution and chemical state of REEs in IAC deposits can be complex. A significant fraction of the REE budget is available via ion-exchange: these REEs are adsorbed on a number of active sites, including: “Broken edge” sites and exposed surface aluminol and silanol groups; hydrophilic silanol surfaces; hydration shells of exchangeable cations; hydrophobic sites on adsorbed organic molecules; isomorphic substitutions (e.g. Al^{3+} for Si^{4+}); and substitution of exchangeable cations (K^+ , Ca^{2+}) (Schoonheydt and Johnston, 2011; Zhou and Keeling 2013). Non-ion-exchange REE forms include colloids deposited as insoluble oxides, hydroxides or polymeric organometallic compounds; mm-to μm -size REE- rich secondary mineral phases such as cerianite-(Ce); and residual refractory magmatic REE- rich minerals (Joussein et al. 2005)

Schoonheydt, R.A. and Johnston, C.T., 2011. 'The surface properties of clay minerals'. In Layered Structures and their Application in Advanced Technologies, EMU Notes in Mineralogy, 11: 337-373.

Zhou, C.H. and Keeling, J., 2013. 'Fundamental and applied research on clay minerals: from climate and environment to nanotechnology'. Applied Clay Science, 74: 3-9.

Joussein, E., Petit, S., Churchman, J., Theng, B., Righi, D. and Delvaux, B., 2005. 'Halloysite clay minerals- a review'. Clay Minerals, 40: 383-426.

We agree entirely, which is why we conducted this study – to measure the structural site of the REE given the range of possible settings, both of the exchangeable REE associated with clays and non-exchangeable REE associated with secondary/refractory phases. We already reviewed some of these papers in the discussion, notably Schoonheydt and Johnston (2011/2013). However, to indicate the complexity of adsorption mechanisms earlier in the manuscript we add a sentence on this in the introduction (Lines 51-54). Our key point here was that carbonates and fluorcarbonates may be present in these profiles as supergene phases, and that they would not be resolved from clay adsorbed REE (by any mechanism) via leaching studies at acid pH. We now clarify this point.

Line 77: “Such studies therefore do not constrain the speciation of the HREE or allow for determination of the behaviour of the economically most significant elements.”

The authors are correct in this assessment. However, this is partly due to the key focus on Ce partitioning within the IACs. This forms the novel research provided in the present study. The focus on Y speciation compared to Nd offers the first look at HREE absorption/complexation within IACs. This point can be better emphasised.

We agree entirely with the referee. We note that Ce partitioning is of scientific interest for its redox behaviour, but not a key focus because Ce is not an economic product of IACs. This statement is pointing out the novelty in our study. We further emphasise the point in the introduction.

Figure 1: The authors need to add the IAC deposits of Philippines. The prospects of N. Vietnam are also not properly shown.

Sanematsu, K. and Watanabe, Y., 2016. Characteristics and genesis of ion adsorption-type rare earth element deposits. Rev. Econ. Geol, 18, pp.55-79.

Padrones, J.T., Imai, A. and Takahashi, R., 2017. Geochemical behavior of rare earth elements in weathered granitic rocks in northern Palawan, Philippines. Resource Geology, 67(3), pp.231-253.

Not all deposits globally were shown because of space constraints. We modify Figure 1 to include deposits in Laos, Philippines and Vietnam. Extra references are included in Line 72-73.

Line 131: There is a sentence missing. I am assuming it is minor 7 Å halloysite (through dehydration of 10 Å halloysite)?

Now corrected. Line 131-134

Line 138: The authors provide no mention of the XRD data for the Ambohimirahavavy complex, but in S2 make a note that the samples are distinct from the Malagasy samples in that 10 Å halloysite is largely absent. They refer to Estrade et al. 2019, but should make a mention of halloysite and other major/minor mineralogy since they report it for Zhiabei (given its importance that is discussed further below).

XRD data for these samples are reported in Estrade et al. 2019. We summarize the mineralogy and add a sentence to clarify clay mineralogy from XRD: Lines 147-149: “These samples were investigated for their mineralogy by Estrade, et al. ³¹ who reported 5 to 50 mass % of kaolinite, with subsidiary, but significant halloysite and minor gibbsite”

We also edit the supplementary figure captions (Fig S3) to acknowledge the presence of minor halloysite-10 Å, and also include SEM images of processed clay fractions (also shown below) in Fig S4 demonstrating dominance of kaolinite in the Malagasy samples we studied.

Figure 3 can be modified to differentiate between kaolinite and halloysite. Changing contrasts in the SEM shows brighter/darker zones easily observable. The zonation is somewhat observed in Fig 3f. Optical microscopy can also show this.

We modify Fig 3 where possible but disagree that we can distinguish kaolinite from halloysite at this scale in BSE images. Halloysite is difficult to resolve in any case (see below). Our XRD and FTIR data suggest it is a minor constituent in these samples, hence the clays in the BSE images are inferred to be dominantly kaolinite. Separated kaolinite fractions also show very few or no grains with tubular morphologies (see SEM images of kaolinite separated for isotopic analyses and experimental work below, although not all halloysite has a tubular morphology). Halloysite is not ubiquitous in the samples we studied – even in the samples used in Ram et al (2019) it is only 22% of the aluminosilicate clay fraction.

Added to that, because halloysite-10 Å ($\text{Al}_2\text{Si}_2\text{O}_5(\text{OH})_4 \cdot 2\text{H}_2\text{O}$) can easily dehydrate to halloysite-7 Å ($\text{Al}_2\text{Si}_2\text{O}_5(\text{OH})_4$, i.e. chemically identical to kaolinite) naturally as well as during sampling/sample prepping, identifying halloysite using conventional mineralogical techniques (XRD, IR, Raman, SEM, EMP, DTA-TG) is ambiguous, especially when mixed with kaolinite, and even more so if kaolinite is disordered. We mention this in the text.

Joussein et al (2005): Because the interlayer water is weakly held, halloysite-(10 Å) can readily and irreversibly dehydrate to give the corresponding halloysite-(7 Å) form. It is therefore very difficult, if

not impossible, to handle halloysite-(10 Å) without inducing some alteration in its hydration state. Only by storing in a sealed container in contact with water, or in a water-saturated atmosphere, could the material be kept fully hydrated (Giese, 1988). Natural or induced dehydration may decrease the cation exchange capacity (Grim, 1968) as well as the ease with which halloysite intercalates organic species (Churchman & Theng, 1984; Joussein et al., submitted).

‘The difference between halloysite-(7 Å) and disordered kaolinite remains very difficult to determine, especially if they are mixed together.’

Figure 4: “The ‘hotspots’ of Y associated with clay minerals occur where no distinct bright REE-rich phases are identified in backscatter and EDS analyses. Element maps for the Malagasy samples with the highest leachable HREE fractions show similar elemental distributions, with Y visibly enriched around the edges of clay minerals.”

The authors need to provide more information here. They only show Y maps with Fe and Mn. The SXRF should also have Ce and La (maybe Nd?) maps which also need to be included. They will show ubiquitous distribution of the LREE which follow a similar trend to the Y. However, they also note higher Y visibly enriched around the edges of clay minerals. Ram et al. 2019 reports that “while the majority of the Y is homogeneously distributed throughout the clay, i.e. likely sorbed as trivalent Y_{3+} ; there is a preferential enrichment in the white phase (consisting mainly of halloysite) relative to the tawny phase (kaolinite).” This is an important distinction with the Malagasy samples, and is worthy of further characterisation to resolve the current data to their association within individual clay fractions. In the case of halloysite, the tubes are highly disordered, ranging in length from 1 μm to 15 μm . Joussein et al. (2005) reported evidence of these tubular morphologies/fibrous structures that are commonly bent and irregular, and could likely be a source of higher adsorption of Y.

Ram, R., Becker, M., Brugger, J., Etschmann, B., Burcher-Jones, C., Howard, D., Kooyman, P.J. and Petersen, J., 2019. Characterisation of a rare earth element-and zirconium-bearing ion-adsorption clay deposit in Madagascar. *Chemical Geology*, 522, pp.93-107.

Joussein, E., Petit, S., Churchman, J., Theng, B., Righi, D. and Delvaux, B., 2005. 'Halloysite clay minerals- a review'. *Clay Minerals*, 40: 383-426.

On the distinction between kaolinite and halloysite: We agree that halloysite is present in some of the Malagasy profiles, but Estrade et al. (2019) show that it is not ubiquitous, and Ram et al.'s data show that halloysite made up less than 22% of the clay concentration in the samples they studied. SEM examination of clay mineral separates (obtained for isotopic study and experimental work, now included as supplementary Fig S4) indicates very little tubular morphology in our Malagasy samples. Because halloysite is largely absent from the Chinese clays (Fig S3) and in Madagascar occupies only a small part of the clay fraction, we focussed mainly on REE adsorption to kaolinite. We have amended the text to acknowledge recent studies (Li et al 2019, Ram et al. 2019, Estrade et al., 2019) that suggest halloysite may be a more efficient adsorber of REE, particularly deeper in the weathering profiles where fine/nanocrystalline and disordered halloysite is inferred to have a higher CEC and SSA than disordered kaolinite.

However, we also note that the Y EXAFS of clays from China and Madagascar are identical, pointing to a similar structural state regardless of which clay it is adsorbed to. With kaolinite being the dominant clay mineral we maintain that our conclusion with respect to kaolinite being the main host of the REE is robust, which is consistent with the recent experimental study of Yang et al 2019 which demonstrated the following: “Compared to halloysite, kaolinite possessed a higher specific surface area (SSA) normalized adsorption capacity towards REEs”. We include a section to this effect in the revised discussion, Lines 394-406.

On SXRF mapping of other elements: Because our primary aim for this study was identifying distribution and coordination of heavy REE (using Y as proxy) we did not acquire SXRF maps for all REE. The incident x-ray excitation energy is chosen to optimise excitation of the elements of interest – one cannot excite efficiently Y (with an excitation energy at 17 keV) at the same time as efficiently exciting Nd (7.1 keV). Nevertheless, limited mapping of Nd, Ce, Fe and Mn from the Chinese samples (by repeating maps multiple times at different x-ray energies) was attempted and qualitatively showed a common uniform distribution in the clays. These maps were measured at lower incident beam energy to optimize for Nd but these were considered unfit for publication due to considerable energy overlaps, swamped detectors and other artifacts. As outlined in the methods of Ram et al, 2019, element mapping of REE other than Y presents challenges due to the overlaps of the X-ray lines with each other and the potential of swamping by other elements (e.g. Mn K overlapping Nd L).

Y K-edge XANES

The authors report: “They compare most closely to XANES of parisite-(Ce) and Y in solution, in which Y occupies 9 and 8-fold coordinated sites, respectively, and existing as 3+ oxidation state. The XANES of the 8-coordinated high symmetry standards (zircon and xenotime) also show Y as 3+ oxidation state. The spectra show a systematic shift in the position and shape of the broad peak around 17104 eV (Feature C), shifting to higher energy with decreasing coordination numbers (from 17104 eV for Y in low-symmetry 11 to 8-fold sites, to 17110 eV for phases with Y in 6-coordinated sites.”

So, am I correct in the assessment that this proves that Y exists within the zircon lattice and hosted within the zircon rather than co-sorption of Zirconium alongside Yttrium? If so, this is an important distinction to be made, as previous SXRF has shown Zr to be ubiquitous with Y in distribution in the IAC's with localised hot spots associated with zircons. So clarity would be appropriate on whether all Y/Zr relationships exist primarily related to Zircons and not otherwise.

The reviewer confuses two things here: 1) XANES spectra from the clay hosted Y fraction (which give similar spectra to that of parisite and solution standards) and 2) Y XANES from relict zircon grains within the samples. Where we discuss the structural state of Y in relict zircon grains this is where Y-hotspots could clearly be correlated with zircon grains identified by EDS. The Y spectra obtained from relict zircons directly match those of Y from crystalline zircon standards (Fig 5a), consistent with 8 fold coordination in the zircon lattice (likely substituting on the Zr-site). This is not the same for the clay-hosted Y XANES, which are comparable to Y in solution and consistent with an 8-fold hydrated coordination sphere. The XANES spectra for clay-adsorbed Y and zircon-hosted Y are clearly different.

We did not perform mapping of Zr XAFS in this study; hence we cannot comment on the speciation of Zr associated with clays or the possibility that Zr may locally be co-adsorbed with Y in the clay fraction, as suggested by Ram et al. We now cite the study of Ram et al. at this point (Line 237-241). This does not detract from our point that the XANES of clay adsorbed Y are clearly distinct from (unleachable) Y held in different coordination states with the zircon lattice of relict zircon grains.

Nd L₃-edge XANES

The Nd XANES shows the Nd exists primarily in +3 oxidation state. Feature B varies significantly from Madagascar and China and is not mentioned properly here. This needs to be explore further. The variable Ce₃₊ to Ce₄₊ ratio is also dependent on Mn/Fe mineralogy which is not reported herein.

This is a good observation, we now note the variation in feature B at 6287 eV between Madagascar and China samples and have changed the Nd XANES results section. The Ce oxidation state has been previously investigated in more detail by Janots et al. and Ram et al. and was not the focus of this study. However, because of the proximity in energy of the X-ray adsorption edges it is impossible to resolve only the Nd L_3 adsorption edge, therefore the Ce adsorption edge always appears in the spectra. These provide insight into the Ce oxidation state, but Ce XAFS was not collected at high resolution as it was not the focus of this study. The Ce^{3+} to Ce^{4+} ratio is indeed correlatable with the Mn/Fe mineralogy (see our Figure 4) – most notably with Mn oxides. This affirms previous studies demonstrating the effect of Mn-Fe oxides in the oxidation of soluble Ce^{3+} and scavenging as Ce^{4+} . We now mention this in Lines 264-272.

Coordination of Y (EXAFS)

Please include models of the localised bonding environment of Y within the spectra itself. This allows for easier readability. This can be done using easy to use software. The primary result suggests that the 8-fold coordinated sites with respect to Y and O.

We now include ball and stick models to show the localised bonding environment of Y for each of the standards below the diagrams of Fig 7 for clarity.

When the authors discuss the aqueous phase of Y (and Nd) do they refer to a leachate solution of this ore, and synthetic solution interchangeably. They provide greater distinction of their conditions in S5 and S6 but will be worth some clarification in the main text.

We always refer to a standard solution of Nd and Y in weak acid (as described in the sample preparation), never a leachate solution of the ore. We clarify this point.

Discussion

A significant portion of the discussion is well established knowledge; the nature of REE_{3+} adsorbed to IACs. See previous papers herein. However, a key distinction is between kaolinite and halloysite. Given the dewatering characteristics of kaolinite vs. halloysite, previous studies have shown greater sorption to halloysite. This needs to be properly reflected within the figure and discussion. The greater incidence of REE as adsorbed fraction is also well established and therefore its greater leachability. What this paper is missing significantly is providing context of the incidence of the REE (which is HREE dominated) between China and Madagascar and other such regolith-hosted clay deposits. For e.g. Ram et al. 2019 highlight the potential for Madagascar to be a viable non-chinese alternative to HREE through comparison of Ce anomaly and total REE+Y concentrations (See Figure below). While the current data show that REE-adsorbed clays from Madagascar and China are direct structural analogues in which the REE occur dominantly as easily exchangeable 8 to 9 coordinated, hydrated, outer sphere, basal surface complexes adsorbed to kaolinite (and possibly halloysite), rather than inner-sphere complexes, or interlayer complexes, the distinction between kaolinite and halloysite is certainly the novelty this paper can provide. Further comparison of these deposits to other deposits with data readily available can provide critical context in the search for critical REE deposits in the weathering environment.

A summary of the clay structure and charge sites is essential for the further discussion and interpretation of the EXAFS data and the interpretation of REE absorption style. With regards to the comparison of total REE with Ce in relation to other deposits, the incidence of the REE is HREE dominated in none of these deposits. The Chinese deposits are only HREE dominated at specific sites (Xu et al., 2017). Ram et al., 2019 used only 4 samples from 2 vertical transects of a site which are atypical of the Ambohimirahavavy complex as a whole (Estrade et al., 2019), and showed only total REE correlations with Ce anomaly, not HREE versus LREE, so it cannot necessarily be claimed that they showed Madagascar is a viable economic alternative for HREE based on such a limited dataset.

While our focus is on the REE adsorption style previous studies have alluded to the mineralogy in more detail. Although halloysite occurs in some regolith profiles of the Ambohimirahavy complex, the more extensive data of Estrade et al., 2019 (105 samples from 4 boreholes, 4 hand dug pits and 2 separate vertical transects through road cuts) show that it is not ubiquitous, and mostly limited to the interior of the caldera in areas inferred to overly SiO₂ undersaturated nepheline syenite. These are indeed the same regoliths studied by Ram et al (one sample from pit A1116 and 3 samples from pit A1153, both of which are right next to PIT1 and PIT2 from Estrade et al. 2019, the latter of which was included in the present study. Estrade et al 2019 demonstrated that regolith mineralogy and REE leaching/adsorption behaviour within the regolith profile is strongly linked to geochemical/mineralogical variations in the bedrock. The lithology and mineralogy of the complex is highly heterogeneous, and this is reflected both in the clay mineralogy (halloysite vs kaolinite) and the leachable REE contents. We clarify this further in the discussion.

Ram et al. demonstrated the presence of halloysite (at lower concentrations than kaolinite) in their samples, but their SXRF mapping shows a broadly uniform distribution of Y across kaolinite rich areas and does not demonstrate preferential adsorption of Y onto halloysite. The referee states that halloysite has greater adsorption potential for the REE and that this has been asserted in a number of recent studies, with studies citing that higher fractions of nanocrystalline halloysite appear to be associated with higher exchangeable REE concentrations (Li et al 2019). However, Schoonheydt and Johnston (2013) as well as Yuan et al (2015) note that the variable charge aluminol surfaces are on the inside of the halloysite tubes, consequently reducing adsorption potential, and Yang et al. 2019 recently demonstrated experimentally that at a given pH and ionic strength kaolinite has a higher adsorption for the REE than halloysite - *'Compared to halloysite, kaolinite possessed a higher specific surface area (SSA) normalized adsorption capacity towards REEs.'* Given the inconsistent reports relating to the absorption capacities of kaolinite versus halloysite in the literature, we disagree that the consensus unequivocally shows higher adsorption of REE to halloysite. Indeed, this subject is of significant importance but beyond the scope of the present study. We stress that the Y EXAFS from the clays in China and Madagascar are identical, demonstrating the adsorption of outer sphere hydrated ions in 8 or 9 fold co-ordination regardless of which clay it is adsorbed to. In every study kaolinite is present in higher amounts than halloysite. While both may be implicated in REE adsorption, the dominance of kaolinite over halloysite in our samples shows that kaolinite is a dominant host for REE in these deposits.

REVIEWERS' COMMENTS:

Reviewer #2 (Remarks to the Author):

Firstly, I want to commend the authors for being highly reciprocating of the critical feedback they received in my assessment.

They have made substantial changes to the manuscript in order to clarify several of my points.

The paper now presents a vital contribution to this field.

I look forward to citing this paper in the future and have provided my reviewer information to be used in the acknowledgement (Rahul Ram).

I have one minor addendum which can be taken as either a note or incorporated within the manuscript:

Chasse et al., 2016 shows that Sc exists both as adsorbed (80%) and within the crystal lattice (20%) of goethite. Given the context of critical metals, and Sc technically considered as part of the REEs, a brief mention of Sc can potentially enhance the conclusion.

Chassé, M., Griffin, W.L., O'Reilly, S.Y. and Calas, G., 2016. Scandium speciation in a world-class lateritic deposit.

REVIEWERS' COMMENTS:

Reviewer #2 (Remarks to the Author):

Firstly, I want to commend the authors for being highly reciprocating of the critical feedback they received in my assessment.

They have made substantial changes to the manuscript in order to clarify several of my points.

The paper now presents a vital contribution to this field.

I look forward to citing this paper in the future and have provided my reviewer information to be used in the acknowledgement (Rahul Ram).

I have one minor addendum which can be taken as either a note or incorporated within the manuscript:

Chasse et al., 2016 shows that Sc exists both as adsorbed (80%) and within the crystal lattice (20%) of goethite. Given the context of critical metals, and Sc technically considered as part of the REEs, a brief mention of Sc can potentially enhance the conclusion.

Chassé, M., Griffin, W.L., O'Reilly, S.Y. and Calas, G., 2016. Scandium speciation in a world-class lateritic deposit.

We thank the reviewer for these positive comments on our revisions, and want to thank the reviewers again for their constructive comments in the first round of reviews.

We agree this is a relevant citation, as it provides another example of the application of X-ray absorption spectroscopy to the speciation of REE in regolith-hosted deposits. We now mention this paper in the introduction.